# ViVidCam: Learning Unconventional Camera Motions from Virtual Synthetic Videos

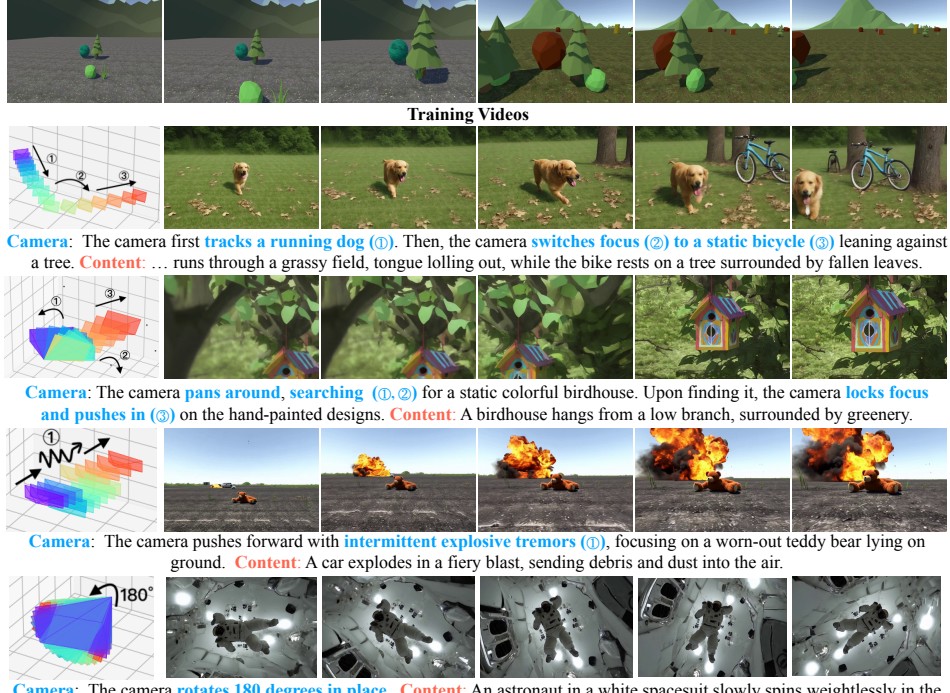

Figure 1: **ViVidCam** learns diverse unconventional camera motions from synthetic videos. The training data (1st row) are simple low-poly 3D scenes rendered in Unity in about *5 seconds* per video. In contrast, the generated results show high visual quality with meaning-driven motions that convey intention (2nd–3rd rows) and more dramatic, unusual motions for artistic effect (4th–5th rows).

## Abstract

Although recent text-to-video generative models are getting more capable of following external camera controls, imposed by either text descriptions or camera trajectories, they still struggle to generalize to unconventional camera motions, which is crucial in creating truly original and artistic videos. The challenge lies in the difficulty of finding sufficient training videos with the intended uncommon camera motions. To address this challenge, we propose ViVidCam, a training paradigm that enables diffusion models to learn complex camera motions from synthetic videos, releasing the reliance on collecting realistic training videos. ViVidCam incorporates multiple disentanglement strategies that isolate camera motion learning from synthetic appearance artifacts, ensuring more robust motion representation and mitigating domain shift. We show that our design synthesizes a wide range of precisely controlled camera motions using surprisingly simple synthetic data. Notably, this synthetic data often consists of basic geometries within a low-poly 3D scene and can be efficiently rendered by engines like Unity. Our video results can be found in https://anonymoususers196.github.io/VividCamDemo/.

## 1 INTRODUCTION

In creative video generation, camera motion is pivotal for conveying intent, enhancing expressivity, and adding artistic value. As a result, recent work in video generation has focused heavily on equipping text-to-video models with camera control. Camera control in video generation is commonly approached via two paradigms: *text-based control,* where motion is described directly in the input prompt (Liu et al., 2024; ArtistT2V.; Google, 2025), and *trajectory-based control*, where explicit 3D motion trajectories are provided as additional conditioning (He et al., 2024; Bahmani et al., 2025; Wang et al., 2025). With sufficient training data labeled with camera motion, both paradigms can effectively reproduce similar motion patterns in generated videos.

However, truly creative video generation demands more than just replicating conventional camera techniques like panning or dollying. It requires inventing stylized, intricate motions (as exemplified by the dramatic impact of the *Dolly Zoom* in Hitchcock's *Vertigo*), or crafting scene-specific movements tailored to expressive content (e.g., tracking an unconventional car race). In such cases, collecting enough training data that embodies these avant-garde or bespoke camera motions is infeasible.

Unfortunately, without enough training data support, neither the text-based nor trajectory-based camera control can generalize well to unseen camera motions. For example, Figure 2 illustrates results from state-of-the-art baseline method (Bahmani et al., 2025): while it can reproduce conventional motions such as a forward push ($1^{st}$ row), it fails on

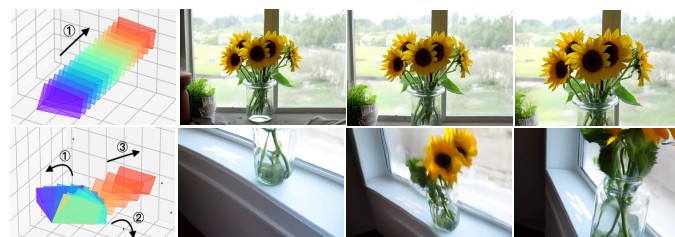

Figure 2: State-of-the-art method (Bahmani et al., 2025) fails to generate unconventional camera motions. More examples are in Appendix D.

more complex, expressive ones ($2^{nd}$ row). In this case, the intended motion was *the camera pans left and right, seeking a sunflower in a glass vase, then locking focus and pushing in upon finding it* (full video: `https://anonymoususers196.github.io/VividCamDemo/`). As shown, the generated videos fail to follow the delicate control of the camera panning left/right process, resulting in videos with large perturbations and losing focus.

In short, there is a fundamental paradox between the data-intensive nature of training generative AI models and the creative nature of camera control. So our question is: Can we enable learning out-of-distribution camera motions without real training data?

In this paper, we explore an alternative solution: instead of collecting real videos with uncommon camera motions, we *generate synthetic videos* with the intended motions as training data for generative models. However, while synthetic videos can cover arbitrary motions, they often exhibit virtual styles that diverge significantly from realistic ones. Directly training models on such videos would seriously degrade generation quality. Of course, such drawbacks could potentially be alleviated by generating super-high-quality, real-like synthetic videos (Shuai et al., 2025), but this would incur tremendous manual efforts and professional expertise to prepare these videos.

Essentially, the problem boils down to a *disentanglement problem*, *i.e.,* separating appearance and camera motion information in synthetic videos and guiding models to learn only the latter. To this end, we present VIVIDCAM, which uses synthetic **Vi**rtual **Vid**eos to fine-tune models for producing correct **Cam**era motions. VIVIDCAM focuses on disentanglement mechanisms. First, inspired by AnimateDiff (Guo et al., 2023), we adopt a dual-adaptation training scheme: we first learn the appearance of synthetic videos through a LoRA and then learn the camera motion; at inference, the appearance LoRA is discarded so the outputs no longer carry undesirable virtual styles. However, this technique has only been used to bridge minor appearance gaps, and is insufficient to resolve the drastic appearance differences between realistic and virtual videos. To further mitigate virtual appearance, VIVIDCAM employs a training recipe with two complementary components: (i) *Data*: we synthesize two sets of videos for training, one without camera motions (to train the appearance LoRA) and one with motions (to train the motion module); and (ii) *Training signals*: we introduce an optical-flow based loss for the motion module, providing appearance-invariant supervision that stabilizes motion learning and strengthens disentanglement. Finally, we use special text prompts to *anchor* virtual appearance, enabling the model to better distinguish virtual from realistic styles.

We find that even when the generated videos are of very low visual quality, comprising only basic geometries rendered in low-poly 3D scenes (top row in Fig. 1), VIVIDCAM can still effectively disentangle the synthetic appearance from motion and generate realistic videos with the camera motion learned from the virtual videos (bottom four rows in Fig. 1). Our experiments demonstrate that models trained with VIVIDCAM can handle a wide variety of complex, compound camera motions while maintaining realistic visual quality comparable to models trained on real footage.

In summary, our main contributions are as follows:

1. We introduce VIVIDCAM, a novel framework for generating realistic videos with diverse camera motions by leveraging synthetic data efficiently rendered from engines like Unity.

2. Despite the significant domain gap, VIVIDCAM effectively mitigates artifacts present in low-quality synthetic videos and focuses on learning their complex camera motions.

3. We demonstrate that our framework can synthesize a wide range of camera motions with precise, consistent control and high visual quality.

## 2 RELATED WORK

### 2.1 VIDEO GENERATIVE MODELS

Early explorations in video generation tasks focused on GANs (Saito et al., 2017; Tulyakov et al., 2018). Recently, leveraging advancements in diffusion models (Ho et al., 2020; Song et al., 2020; Peebles & Xie, 2023; Ma et al., 2024a), video generative models have rapidly evolved. Early diffusion-based video generative models originated from adapting existing image generative models by incorporating additional motion modules (Guo et al., 2023; Blattmann et al., 2023b; Gu et al., 2023; Singer et al., 2022; Wu et al., 2023). More recently, many end-to-end video generation models have achieved superior results in terms of video quality, resolution, and duration (Blattmann et al., 2023a; Hong et al., 2022; Yang et al., 2024b; Ma et al., 2024b). For conditional video generation, various works have explored using text or images to guide the content in generated videos. Rapid progress in video generation has led to several groundbreaking works (OpenAI, 2024; Wang et al., 2024a; Kuaishou, 2025; Google, 2025).

### 2.2 CAMERA CONTROL IN VIDEO GENERATION

Recently, a line of research has focused on enhancing the controllability of video generative models (Sun et al., 2024; Zhou et al., 2025; Peng et al., 2024). An important aspect is managing camera motion (Zhang et al., 2024; Xu et al., 2024a; Hou et al., 2024; Ling et al., 2024). Early works learned simple, fixed movements (e.g., zooming, panning) from reference videos (Guo et al., 2023; Blattmann et al., 2023a), while later methods conditioned generation on input trajectories (Xu et al., 2024b; Yang et al., 2024a), representing cameras through camera matrices (Wang et al., 2024c) or Plücker embedding (He et al., 2024; Bahmani et al., 2025). These approaches, however, depend on large annotated datasets, which are scarce and offer only limited motion diversity. Recent works (He et al., 2025; Yu et al., 2025; Wang et al., 2025) have devoted significant effort to constructing and curating large-scale realistic video datasets with camera trajectory annotations.

### 2.3 IMPROVING VIDEO GENERATION MODELS USING SYNTHETIC DATA

Training on realistic video datasets often faces limitations, such as the absence of camera motion annotations and the limited diversity of motion patterns (He et al., 2024; Wang et al., 2024c). As such, several studies turned to synthetic datasets, many of which focus on multi-view generation (Bai et al., 2024) and human animation (Black et al., 2023; Wang et al., 2024b; Yang et al., 2023). For camera motion editing, recent work has synthesized new training videos by modifying camera trajectories in existing sequences (Bai et al., 2025), though these efforts are generally restricted to simple motions. For camera control in video generation, recent approaches generate training videos with explicit camera trajectories in virtual scenes rendered by 3D engines (Fu et al., 2024; Shuai et al., 2025). However, these methods typically demand substantial manual effort to generate complex and diverse scenes and objects. In contrast, our work leverages simple, low-poly 3D environments, enabling diverse camera motions without reliance on large-scale annotation or labor-intensive scene design.

Figure 3: VIVIDCAM. We first render initial scene using publicly available assets. Then, we render videos with and without camera motion. These videos are used to train the camera and appearance modules, generating videos with desired camera motions. Details of training are in Sec. 4.3.

# 3 PRELIMINARIES

We first provide a brief overview of text-to-video diffusion models, which serve as the base model of our work. To train a diffusion model, we first generate a set of corrupted videos, denoted as $\boldsymbol{X}_{1:T}$, by adding progressively increasing Gaussian noises $\boldsymbol{\epsilon}_{1:T}$, to the clean video, $\boldsymbol{X}_0$. The diffusion model then learns to predict the additive noise and denoise noisy videos into cleaner videos. We focus on two types of diffusion models with slightly different training objectives.

**Text-Based Control Only.** Standard text-to-video diffusion models condition the denoising process only on a text input, denoted as $\boldsymbol{c}$. The training loss can be written as

$$\mathcal{L}(\boldsymbol{\theta}_d) = \mathbb{E}_{\boldsymbol{X}_0, t, \boldsymbol{\epsilon}_t}[\boldsymbol{\epsilon}_t - \hat{\boldsymbol{\epsilon}}_{\boldsymbol{\theta}_d}(\boldsymbol{X}_t, \boldsymbol{c}, t)], \tag{1}$$

where $\boldsymbol{\theta}_d$ denotes the parameters of the noise predictor. During inference, clean videos are progressively denoised from pure-noise videos using the trained noise predictor conditional on text $\boldsymbol{c}$. We denote the generation process as $\boldsymbol{g}_{\boldsymbol{\theta}_d}(\boldsymbol{c})$.

**Trajectory-Based Control.** Recent works control camera poses using trajectories represented as Plücker embeddings $\boldsymbol{p}$, which are encoded via $E_{\boldsymbol{\theta}_e}(\boldsymbol{p})$ with parameters $\boldsymbol{\theta}_e$. The training loss becomes

$$\mathcal{L}(\boldsymbol{\theta}_d, \boldsymbol{\theta}_e) = \mathbb{E}_{\boldsymbol{X}_0, t, \boldsymbol{\epsilon}_t}[\boldsymbol{\epsilon}_t - \hat{\boldsymbol{\epsilon}}_{\boldsymbol{\theta}_d}(\boldsymbol{X}_t, \boldsymbol{c}, E_{\boldsymbol{\theta}_e}(\boldsymbol{p}), t)]. \tag{2}$$

The inference generation process becomes $\boldsymbol{g}_{\boldsymbol{\theta}_d}(\boldsymbol{c}, E_{\boldsymbol{\theta}_e}(\boldsymbol{p}))$.

# 4 VIVIDCAM

In this section, we first formulate the research problem of synthesizing unconventional camera motions. Then, we introduce **VIVIDCAM**, a framework to generate them leveraging completely virtual synthetic videos. The overall pipeline is shown in Fig. 3.

## 4.1 PROBLEM FORMULATION

We aim to fine-tune pre-trained text-to-video diffusion models to follow certain camera motions, which can be unconventional so the pre-trained models do not generalize well on them.

Specifically, we consider two camera control paradigms. For **text-based control**, the camera motion instructions are specified as additional input prompts, denoted as $\boldsymbol{c}_m$, such as *'The camera pans around, searching for a bird.'* We fine-tune a *text-only diffusion model*, parameterized as $\boldsymbol{\theta}_d$, to integrate the additional camera instructions, so the generation process becomes $\boldsymbol{g}_{\boldsymbol{\theta}_d + \Delta\boldsymbol{\theta}_{cd}}(\boldsymbol{c}_m \oplus \boldsymbol{c})$, where $\oplus$ denotes text concatenation, and $\Delta\boldsymbol{\theta}_{cd}$ denotes the fine-tuning weight difference.

For **trajectory-based control**, the camera motions are in the form of out-of-distribution camera trajectories $\boldsymbol{p}$. We fine-tune only the *trajectory encoder* of a text-to-video diffusion model, since the encoder already provides basic camera control abilities (Bahmani et al., 2025). The adapted generation process becomes $\boldsymbol{g}_{\boldsymbol{\theta}_d}(\boldsymbol{c}, E_{\boldsymbol{\theta}_e + \Delta\boldsymbol{\theta}_{ce}}(\boldsymbol{p}))$, where $\Delta\boldsymbol{\theta}_{ce}$ are fine-tuning weight difference (only the trajectory encoder weights are updated; the diffusion model weights are kept frozen).

Since the camera motions are unconventional, realistic video datasets often lack sufficient examples for training. We propose to leverage synthetic videos rendered by a physics engine, which allows for arbitrary and diverse camera trajectories. As such, we consider two research questions:

• What types of synthetic videos are most effective for enhancing camera motion learning?

• How to ensure the model learns camera motion independently of the virtual video's appearance?

To answer these questions, we first detail the process for rendering training videos in Sec. 4.2. Then, we present our pipeline for disentangling camera motion from artificial appearances in Sec. 4.3.

## 4.2 RENDER TRAINING VIDEOS

Our first step is to prepare synthetic videos for camera motion training. Leveraging the rendering engine, we are able to generate arbitrary camera motions within a virtual scene. Note that while it is possible to include a variety of synthetic objects in the scene, we focus on constructing the scene with **minimal** numbers and categories of objects to reduce human efforts. Below, we describe the key details of the rendering process. More details can be found in Appendix A.

**The rendered scene.** All synthetic videos are rendered in a low-poly 3D scene using Unity. The scene consists of a *background*, *floor*, and *objects*. Examples of these elements are shown in Fig. 3. Notably, these elements are created using basic geometries. When preparing to synthesize videos, we first randomly sample a background, floor texture, and arbitrarily determine the positions of the objects. Fig. 3 also illustrates an example of the initial settings of the scene.

**Videos with camera motions.** Next, we define the camera motion for the videos. Unity allows users to specify camera movement through code, enabling the simulation of arbitrary camera motions in the scene with just a few lines of code. We consider diverse simple and complex motions listed in Table 1. We denote these synthetic videos with camera motions as $\mathcal{X}_c$ ($c$ stands for 'camera').

**Videos without camera motions.** VIVIDCAM also requires a set of videos without camera motions to aid the disentanglement between appearance and camera motion. For this purpose, as shown in Fig. 3, we synthesize another set of training videos, denoted as $\mathcal{X}_a$ ($a$ stands for 'appearance'), which consist of identical appearance styles but with a static camera.

## 4.3 DUAL ADAPTATION TRAINING SCHEME

Next, we introduce the pipeline to learn camera motions without introducing synthetic side effects by leveraging the rendered data $\mathcal{X}_c$ and $\mathcal{X}_a$. Prior work suggests that a LoRA module trained specifically to capture the appearance of videos can help mitigate the domain gap in realistic video training datasets (Guo et al., 2023). Motivated by this insight, we investigate whether similar techniques can be applied to absorb synthetic artifacts in *fully virtual-style videos*. Our training involves two steps.

• **Step 1: Appearance Adaptation.** We first train a LoRA, called the *appearance LoRA*, to model the visual characteristics of synthetic scenes without entangling motion, using the static videos $\mathcal{X}_a$ that contain no camera movement. The training objective is thus formulated as

$$
\begin{aligned}
\textbf{Text-Based Control:} \quad & \mathcal{L}_1(\Delta\boldsymbol{\theta}_a) = \mathbb{E}_{\boldsymbol{X}_0\sim\mathcal{X}_a,t,\boldsymbol{\epsilon}_t}[\boldsymbol{\epsilon}_t - \hat{\boldsymbol{\epsilon}}_{\boldsymbol{\theta}_d+\Delta\boldsymbol{\theta}_a}(\boldsymbol{X}_t,\boldsymbol{c},t)], \\
\textbf{Trajectory-Based Control:} \quad & \mathcal{L}_1(\Delta\boldsymbol{\theta}_a) = \mathbb{E}_{\boldsymbol{X}_0\sim\mathcal{X}_a,t,\boldsymbol{\epsilon}_t}[\boldsymbol{\epsilon}_t - \hat{\boldsymbol{\epsilon}}_{\boldsymbol{\theta}_d+\Delta\boldsymbol{\theta}_a}(\boldsymbol{X}_t,\boldsymbol{c},E_{\boldsymbol{\theta}_e}(\boldsymbol{p}_0),t)],
\end{aligned}
\tag{3}
$$

where the $\Delta\boldsymbol{\theta}_a$ denote the appearance LoRA parameters, $\boldsymbol{c}$ only contains scene descriptions (and no camera motion descriptions), and $\boldsymbol{p}_0$ denotes a static camera trajectory.

• **Step 2: Camera Control Learning.** We then further fine-tune the models on the new camera motion, in the presence of the trained appearance LoRA (which is kept frozen), using the dataset $\mathcal{X}_c$. The training objective consists of two components. The first is the standard diffusion loss:

$$
\begin{aligned}
\textbf{Text-Based Control:} \quad & \mathcal{L}_2(\Delta\boldsymbol{\theta}_{cd}) = \mathbb{E}_{\boldsymbol{X}_0\sim\mathcal{X}_c,t,\boldsymbol{\epsilon}_t}[\boldsymbol{\epsilon}_t - \hat{\boldsymbol{\epsilon}}_{\boldsymbol{\theta}_d+\Delta\boldsymbol{\theta}_a+\Delta\boldsymbol{\theta}_{cd}}(\boldsymbol{X}_t,\boldsymbol{c}_m\oplus\boldsymbol{c},t)], \\
\textbf{Trajectory-Based Control:} \quad & \mathcal{L}_2(\Delta\boldsymbol{\theta}_{ce}) = \mathbb{E}_{\boldsymbol{X}_0\sim\mathcal{X}_c,t,\boldsymbol{\epsilon}_t}[\boldsymbol{\epsilon}_t - \hat{\boldsymbol{\epsilon}}_{\boldsymbol{\theta}_d+\Delta\boldsymbol{\theta}_a}(\boldsymbol{X}_t,\boldsymbol{c},E_{\boldsymbol{\theta}_e+\Delta\boldsymbol{\theta}_{ce}}(\boldsymbol{p}),t)],
\end{aligned}
\tag{4}
$$

where $\boldsymbol{c}_m$ is the camera motion description. $\Delta\boldsymbol{\theta}_{cd}$ is a LoRA module only used for text-based model. For trajectory-based control, instead of LoRA fine-tuning, we perform full fine-tuning on the camera encoder, denoted by $\Delta\boldsymbol{\theta}_{ce}$.

In addition, we introduce an *optical-flow-based loss* to encourage camera motion learning by aligning frame-to-frame differences between predictions and ground truth:

$$
\mathcal{L}_{\text{flow}} = \frac{1}{K-1}\sum_{k=1}^{K-1}\left\|(\hat{\boldsymbol{X}}_0^{(k+1)} - \hat{\boldsymbol{X}}_0^{(k)}) - (\boldsymbol{X}_0^{(k+1)} - \boldsymbol{X}_0^{(k)})\right\|_1,
\tag{5}
$$

Table 1: Summary of camera motion categories. We classify them into three categories based on their complexity. These motions are derived from realistic applications in everyday videos and filmmaking.

| Category | | Examples | Videos/Filmmaking Applications |
|---|---|---|---|
| **Simple** | | Push in, Tilt up | ❏ Emphasize an object |
| **Composed** | | Push in → Truck left | ❏ Emphasize, then reveal surroundings |
| **Complex** | Expressive | Seek object (pan/tilt search)
Switch focus between objects
Orbit shot
Handheld shake | ❏ Simulate searching for a target
❏ Highlight relational dynamics
❏ Showcase all sides of objects
❏ Simulate instability or realism |
| | Stylized | Dolly zoom
Explosive shake
Camera rotation (90/180°) | ❏ Create dramatic perspective shift
❏ Convey impact or chaos
❏ Disorient viewer, mark transition |

where $\hat{X}_0$ is the predicted video reconstructed from the noisy input $X_t$; $(k)$ denotes the frame index. The final training loss in step 2 is therefore $\mathcal{L} = \mathcal{L}_2 + \lambda \mathcal{L}_{\text{flow}}$, where $\lambda > 0$ balances two losses.

It is important to note that the camera control adaptation is performed on different model components for text- and trajectory-based methods. For text-based control, the camera LoRA is performed on the diffusion model weights, superimposed on top of the appearance LoRA. For trajectory-based control, the adaptation is performed on the trajectory encoder, which is separated from the appearance LoRA on the diffusion model, because the former is responsible for processing the trajectory information.

The fundamental idea behind dual adaptation is that since the appearance LoRA already learns the virtual appearance information, the camera control learning no longer needs to learn the same information, and can focus on what the appearance LoRA does not learn – camera motion.

• *Inference.* After the training, we only deploy camera modules $\Delta\boldsymbol{\theta}_{cd}$ and $\Delta\boldsymbol{\theta}_{ce}$ during inference, while the appearance LoRA $\Delta\boldsymbol{\theta}_a$ is discarded. This would largely remove the undesirable synthetic appearance acquired during training. Specifically, given an input text prompt $\boldsymbol{c}$ with camera instruction $\boldsymbol{c}_m$ or camera pose $\boldsymbol{p}$, the video can be synthesized using $g_{\boldsymbol{\theta}_d + \Delta\boldsymbol{\theta}_{cd}}(\boldsymbol{c}_m \oplus \boldsymbol{c})$ for text-based model or $g_{\boldsymbol{\theta}_d}(\boldsymbol{c}, E_{\boldsymbol{\theta}_e + \Delta\boldsymbol{\theta}_{ce}}(\boldsymbol{p}))$ for trajectory-based model, respectively. A more detailed description and examples of the training and inference prompts can be found in Appendix B.

**Style-aligned Prompt.** While the appearance LoRA helps address domain gaps, we observe that relying on it alone still introduces synthetic artifacts (see Sec. 5.4). To further disentangle the appearance and camera motion learning, during both training stages, which are trained on virtual videos, we append a virtual indicator to the input text prompt, $\boldsymbol{c}$, in the form of *'In this low-poly* <VIRTUAL> *scene.'* This would help the model differentiate the virtual style. During inference, this virtual indicator is dropped, which further removes the virtual appearance quality.

## 5 EXPERIMENTS

In this section, we evaluate VIVIDCAM under various camera motions and compare it with state-of-the-art methods. We focus on the following questions: ❶ What types of camera motions can VIVIDCAM generate? ❷ Can VIVIDCAM provide precise camera control? ❸ Despite being trained exclusively on low-poly synthetic videos, can VIVIDCAM synthesize high-quality realistic videos?

### 5.1 EXPERIMENT SETTINGS

**Implementation.** For all experiments, we use `CogVideoX-5B` (Yang et al., 2024b) as the base model. For *text-based control*, the LoRA rank is 128 and 512 for appearance and camera learning, respectively. Notably, different camera motion types are trained using a single camera LoRA module. For *trajectory-based control*, we fine-tune the `AC3D` model built on `CogVideoX-5B`. Similarly, a single trajectory encoder is trained to handle multiple motion types. More details are in Appendix C.

**Camera Motions.** As shown in Table 1, we systematically consider three broad categories of camera motions: *simple*, *composed*, and *complex* movements. *Simple* camera movements refer to basic motions in six different directions. *Composed* movements combine two simple motions. Specifically,

Table 2: Camera pose precision measurements. The best TransErr and RotErr values are in **bold**, and the second-best are underlined.

| | Simple Motion | | | Composed Motion | | | Complex Motion | | |
|---|---|---|---|---|---|---|---|---|---|
| | TransErr ↓ | RotErr ↓ | FVD ↓ | TransErr ↓ | RotErr ↓ | FVD ↓ | TransErr ↓ | RotErr ↓ | FVD ↓ |
| CAMERACTRL (He et al., 2024) | 0.3578 | 0.1358 | 2577.35 | 0.3835 | 0.1989 | 2000.37 | 0.5753 | 0.7042 | 1503.76 |
| COGVIDEOX (Yang et al., 2024b) | 0.3113 | **0.0297** | 1781.90 | 0.5107 | **0.1338** | 2168.99 | 0.4327 | 0.5067 | 1488.36 |
| AC3D (Bahmani et al., 2025) | 0.2639 | 0.0973 | 1996.32 | 0.4389 | 0.1958 | 2241.88 | 0.4271 | 0.5864 | 1719.82 |
| VIVIDCAM -COG | **0.1704** | 0.0407 | 1808.30 | 0.2208 | 0.1593 | 2162.60 | 0.4011 | 0.5013 | 1866.40 |
| VIVIDCAM -AC3D | 0.2502 | 0.1162 | 2007.15 | **0.1908** | 0.1968 | 2280.71 | **0.3376** | **0.3619** | 1721.45 |

Table 3: Human study results. The best scores are shown in **bold**, and the second-best are underlined.

| | Simple Motion | | Composed Motion | | Complex Motion | |
|---|---|---|---|---|---|---|
| | Action Correctness ↑ | Realism ↑ | Action Correctness ↑ | Realism ↑ | Action Correctness ↑ | Realism ↑ |
| CAMERACTRL (He et al., 2024) | 0.79 | 0.68 | 0.90 | 0.74 | 0.62 | 0.60 |
| COGVIDEOX (Yang et al., 2024b) | 0.80 | 0.74 | 0.53 | 0.65 | 0.61 | **0.68** |
| AC3D (Bahmani et al., 2025) | 0.74 | 0.72 | 0.75 | 0.66 | 0.68 | 0.65 |
| VIVIDCAM -COG | **0.86** | 0.77 | **0.93** | 0.74 | 0.77 | **0.68** |
| VIVIDCAM -AC3D | 0.74 | **0.78** | 0.90 | **0.78** | **0.81** | 0.65 |

when motion 1 is combined with motion 2, the first half of the video follows motion 1, and the second half motion 2. We consider combinations of {push in, pull out} × {truck left, truck right}, resulting in four combinations. Beyond these, *complex* camera motions include more unconventional practices, divided into *expressive* motions that convey semantic meaning (e.g., seeking objects, handheld shake) and *stylized* motions that create dramatic visual effects (e.g., explosive shake, camera rotation). All of these camera motions are motivated by realistic applications in everyday videos and filmmaking.

## 5.2 QUALITATIVE RESULTS

We present qualitative results of text-based control in Fig. 4. Please see Appendix D for trajectory-based methods and comparisons with existing methods. We highlight the following three features:

• **Stable camera motion.** We observe that the generated videos exhibit stable camera motion. For example, in the first row, the camera steadily pushes forward at a constant speed, as evidenced by the predictable changes in the sizes and positions of objects, such as the stone annotated in the frames.

• **Ability to handle dynamic objects.** VIVIDCAM can generate precise camera motions for both static and dynamic objects. For example, in rows 1, 2, 5, and 6, we demonstrate that VIVIDCAM can synthesize high-quality moving objects, such as children, birds, and fire effects.

• **Mastery in unconventional camera motions.** VIVIDCAM can synthesize unconventional camera motions, many of which semantically depict story-like camera shots. For example, in row 4, VIVID-CAM simulates a common scenario where a person looks around for bread and locks focus once they find it. This capability suggests broad applications beyond simple camera movement controls. Additionally, we note that some delicate motions, like shaking in row 6, are difficult to convey through static images. To illustrate this, we annotate the shaking frames to highlight the blur and camera motion. We encourage readers to explore our vivid video results on our anonymous web page.

## 5.3 QUANTITATIVE EVALUATIONS

In this section, we quantitatively compare our framework with existing models and methods. We compare our work with the following representative baselines:

• CAMERACTRL (He et al., 2024) enables trajectory-based camera control for U-Net diffusion models via Plücker embedding. In experiments, we provide both prompts and camera poses as inputs.

• COGVIDEOX (Yang et al., 2024b) demonstrates text-based camera motion control, likely due to exposure to relative motion data during pre-training. In experiments, we provide prompts that combine both camera motion instructions and content descriptions as inputs.

• AC3D (Bahmani et al., 2025) achieves trajectory-based camera control using ControlNet (Zhang et al., 2023) on DiT (Peebles & Xie, 2023) based models. Similar to CAMERACTRL, we provide the same trajectory and prompt description as input.

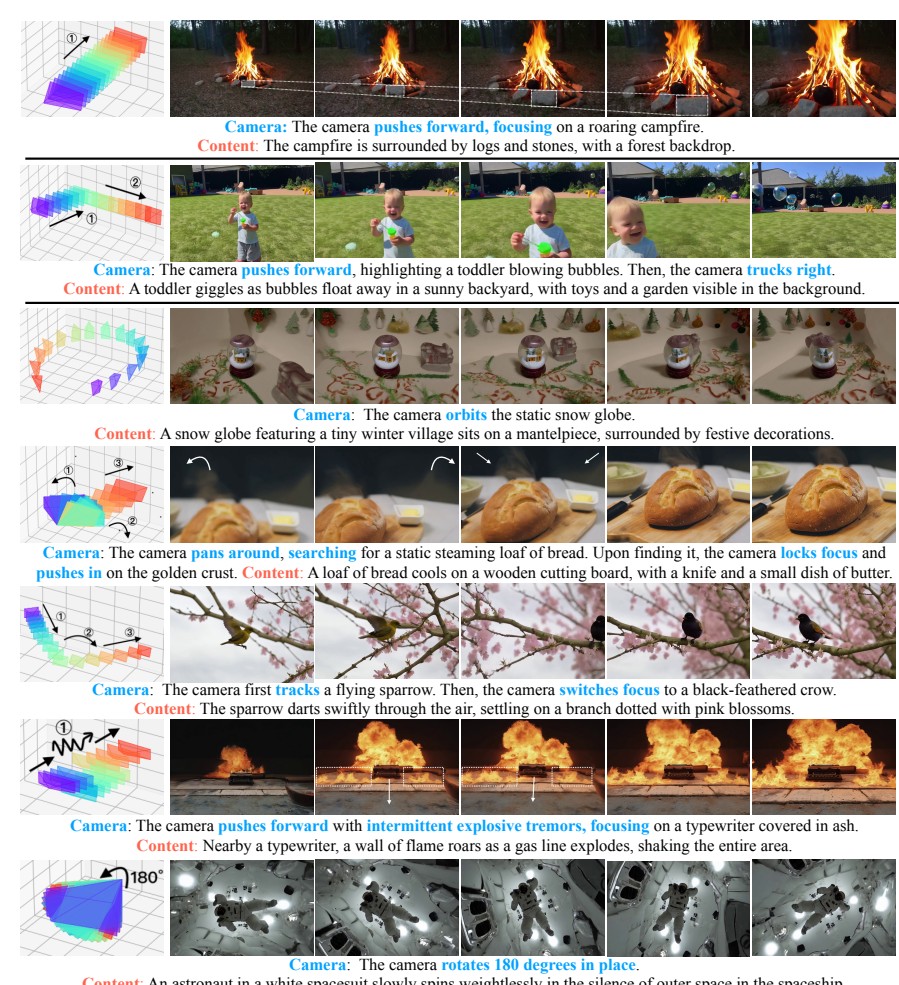

Figure 4: Qualitative results of diverse camera motions. From top to down: ❶ Push forward; ❷ Push forward, then truck right; ❸ Orbit shot; ❹ Pan around, then focus on one object; ❺ Switch focus between objects; ❻ Camera shaking; ❼ Camera rotating. The three panels correspond to simple, composed, and complex camera motions. Note that some complex camera motions are difficult to demonstrate through images; please refer to the videos on our anonymous webpage https://anonymoususers196.github.io/VividCamDemo/ for better visual results.

For all methods, we use the official implementations and checkpoints from their repositories. We conduct experiments across three levels of motion: *simple*, *composed*, and *complex*. Each category consists of 100 input prompts. Details of the prompts can be found in Appendix B.

**Evaluation Metrics.** Following prior work (Cheong et al., 2024; He et al., 2024), we use *FVD* (Unterthiner et al., 2019) to assess visual quality, and report *TransErr* and *RotErr* to evaluate camera action accuracy. However, in the text-based control setting, the model does not have access to the ground-truth trajectory, making these metrics potentially unfair. To address this, we conduct a *human study* to evaluate both the correctness of camera actions and the realism of the generated videos. We report *Action Correctness* and *Realism* scores from 88 participants (see Appendix E for details).

**Automated Evaluation Results.** We present the automated metric results in Table 2. Our text-based and trajectory-based methods are denoted as VIVIDCAM-COG and VIVIDCAM-AC3D, respectively. As shown, our method demonstrates strong camera motion precision across diverse motion categories, as indicated by the low TransErr and RotErr values. Additionally, we highlight that our method preserves video quality, as evidenced by the small FVD difference compared to the vanilla COGVIDEOX and AC3D. Notably, the original CogVideoX achieves good RotErr performance for simple and composed motions. We find this is because such motions involve minimal camera rotation, whereas vanilla CogVideoX typically produces videos with imprecise camera translation (high

TransErr) and lacks camera rotation altogether. The following human evaluation further highlights its limitations, and we provide additional discussion of this phenomenon in Appendix D.

**Human study results.** We present human study results in Table 3. We observe that VIVIDCAM-COG consistently generates more precise camera motion compared to the baselines across all categories of camera motion, and VIVIDCAM-AC3D shows advantages on more complex motions. Additionally, VIVIDCAM produces high-quality, realistic-style videos, as evidenced by the comparable realism scores given by participants.

## 5.4 ABLATION STUDY

In this section, We conduct ablation studies to examine two key design choices: ❶ **Algorithm side:** Is it necessary to incorporate appearance LoRA and style-aligned prompts? ❷ **Data side:** Does including realistic videos in the training data improve performance?

**The role of appearance adaptation and style-aligned prompts.** We first examine how appearance LoRA and style-aligned prompt design help mitigate the negative effects of synthetic appearance. For ablation, we train one model without the appearance adaptation step. In parallel, we train another model in which the style-aligned prompt is omitted from the input text during training.

We conduct a human evaluation and show results in Table 4. Overall, we find the absence of appearance adaptation and style-aligned prompts leads to a clear decline in the model's ability to synthesize realistic videos. As shown in Figure 5, without appearance adaptation and style-aligned prompts, the generated appearance closely resembles the synthetic data, resulting in significantly degraded video quality (e.g., synthesized texture in $1^{st}$ row and distorted glasses in the $2^{nd}$ row). In contrast, our full model configuration effectively mitigates these visual artifacts, producing clean, realistic videos with the desired camera motion.

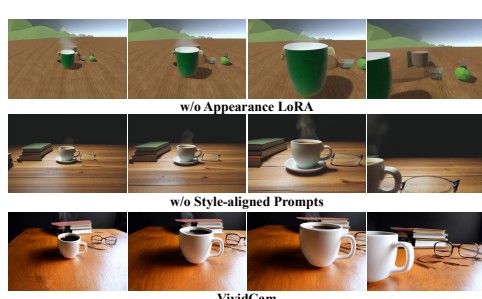

Figure 5: Visual examples illustrating the effects of appearance LoRA and style-aligned prompts.

Table 4: Effects of appearance LoRA (A-LoRA) and style-aligned prompts (S-Prompt).

|  | Action Correctness | Realism |
|---|---|---|
| w/o A-LoRA | 0.75 | 0.60 |
| w/o S-Prompt | **0.93** | 0.65 |
| VIVIDCAM | **0.93** | **0.74** |

**Add realistic videos in training data.** Second, we examine whether adding realistic videos improves training. We use the RealEstate10K (Zhou et al., 2018) dataset, which provides annotated camera trajectories. Thus, experiments are conducted in the trajectory-based camera control setting. The training set includes 500 synthetic and 500 realistic videos. Results in Table 5 indicate that using only virtual training data performs comparably to mixing in realistic videos. We hypothesize that realistic data offers little benefit due to the limited diversity of camera motions in RealEstate10K. Given the extra human effort required to collect large-scale realistic data, VIVIDCAM offers a cost-efficient training paradigm.

Table 5: Effects of realistic data in training.

|  | Action Correctness | Realism |
|---|---|---|
| Mixed data | 0.88 | 0.75 |
| Virtual data | 0.91 | 0.72 |

## 6 CONCLUSION

In this paper, we propose **VIVIDCAM**, which uses synthetic **Vi**rtual **Vid**eos to fine-tune video generation models to generate correct **Cam**era motions. Notably, we show that *synthetic videos do not need to be realistic at all*. In fact, VIVIDCAM shows that video diffusion models can effectively learn camera motion from surprisingly simple synthetic data, often comprising basic geometries rendered in low-poly scenes. Experiments show that models trained with VIVIDCAM can master various compound and complex camera motions, while maintaining a level of realism comparable to baselines trained on real footage. Ultimately, our work offers an efficient approach to synthesizing realistic videos with precise camera motion control, especially for unconventional motions.

**Reproducibility Statement.** For reproducibility, we detail the processes of synthetic data creation, algorithm implementation, and experimental setup. Specifically, we describe the process of rendering training videos in Sec. 4.2 and Appendix A; the implementation of algorithms and hyperparameter settings in Sec. 5.1 and Appendix C; and the experimental setup in Sec. 5.

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

## A    RENDERING TRAINING VIDEOS USING UNITY

In this work, all training videos are synthesized using Unity. As introduced in Sec. 4.2, for each camera motion synthesis, we first prepare the scene and then render the video with and without camera motion.

**Scene creation.** Each scene consists of a background, a floor texture, and objects. These elements are randomly determined during scene creation. Specifically, we first randomly select the categories of the background and floor texture. The background options include {"sky", "far mountains", "closer mountains", "both mountains"}. The "sky" refers to the default background in Unity. The "far mountains" and "closer mountains" are publicly available background assets that depict mountains at different distances, respectively. The "both mountains" option includes both of the previous mountain backgrounds. The floor texture options include "brick and stone floor", "black sand ground", "green grassland", "brown ground", "yellow grassland", "light green grassland". These textures are also publicly available assets in Unity. For the objects, we randomly place both static and moving objects in the scene. The static objects include "tree", "bush", "grass", while the moving objects include "sphere", "cube", "polygon", "cylinder". Notably, all objects are created using basic geometric shapes and do not require specific human effort for design. Please refer to the example training videos on our anonymous website `https://anonymoususers196.github.io/VividCamDemo/` for a better understanding of their visual appearance.

**Video rendering.** After creating the scene, we render the videos both with and without camera motion. The videos without camera motion are generated by randomly determining the camera's coordinates and pose, then fixing the camera in place while recording the video. For the videos with camera motion, we first define the camera movements using a short script (typically no more than 10 lines of code). Based on this script, the rendered video incorporates the specified camera motions. We note that the camera motion script can be effectively written by GPT given natural language instructions (*e.g.,* "I want to write a Unity C# code depicting a camera first push forward, then truck left.")

## B    DETAILS OF TEXT PROMPTS

Our training and inference processes rely on different categories of text prompts. In this section, we provide a detailed discussion of the prompts used. Generally, two categories of prompts are employed: (1) scene-only prompts $c$, used for appearance LoRA learning, and (2) composite prompts $(c_m \oplus c)$, which combine camera instructions with scene descriptions for learning camera control in the text-based setting. Additionally, we provide examples of prompts used during inference.

**Scene-Only Prompts for Appearance LoRA Training.** During appearance LoRA training, we constrain the LoRA to learn only the appearance style. Therefore, the training prompt at this stage includes only a description of the rendered scene, specifying objects and environmental details. For example: *"Content: There are small plants and geometries on the light green grassland."* Additionally, as described in Sec. 4.3, we incorporate a style-aligned prompt to help bridge domain gaps during appearance LoRA training. This prompt acts as a virtual indicator of the target style. With this addition, the complete training prompt $c$ becomes, for example: *"Content: In this low-poly 3D `<VIRTUAL>` scene, there are small plants and geometries on the light green grassland."*

**Composite Prompts for Text-Based Camera Control.** For text-based camera control, we freeze the appearance LoRA and train a separate camera LoRA. At this stage, the training prompt includes both camera movement instructions $c_m$ and scene descriptions $c$. The camera component guides the camera LoRA to learn appropriate motion patterns, while the scene description ensures consistent content generation. For example: *"Camera: The camera pushes forward, focusing on a moving sphere. Then the camera trucks left. | Content: In this low-poly 3D `<VIRTUAL>` scene, there is a moving sphere. There are also small plants and geometries on the black sand ground."* It is worth noting that for trajectory-based camera control, we use only the scene description $c$, rather than composite prompts $(c_m \oplus c)$, since the camera condition is provided directly by the trajectory input $p$.

**Prompts at Inference Time.** During inference, we use prompts similar to those employed during camera control training, with the exception that the virtual style indicator is omitted. Below are example prompts for both text-based and trajectory-based control: Text-based: *"Camera: The*

|  | | Value |
|---|---|---|
| **Appearance LoRA** | Learning rate | 1e-4 |
|  | Rank | 128 |
|  | Scheduler | Cosine with Restarts |
|  | Warm up steps | 400 |
|  | Optimizer | adamw |
|  | $\beta_1$ | 0.9 |
|  | $\beta_2$ | 0.95 |
| **Camera LoRA** | Learning rate | 3e-4 |
|  | Rank | 512 |
|  | Scheduler | Cosine with Restarts |
|  | Warm up steps | 400 |
|  | Optimizer | adamw |
|  | $\beta_1$ | 0.9 |
|  | $\beta_2$ | 0.95 |
| **Trajectory Encoder** | Learning rate | 1e-4 |
|  | Scheduler | Cosine with Restarts |
|  | Warm up steps | 250 |
|  | Optimizer | adamw |
|  | $\beta_1$ | 0.9 |
|  | $\beta_2$ | 0.95 |

Table 6: Hyperparameter settings.

*camera pushes forward, focusing on a static steaming coffee cup. Then the camera trucks right. | Content: A steaming coffee cup rests on a wooden table beside a stack of books and a pair of glasses."* Trajectory-based: *"Content: A steaming coffee cup rests on a wooden table beside a stack of books and a pair of glasses."*

## C  IMPLEMENTATION DETAILS

For all experiments, we use `CogVideoX-5B` (Yang et al., 2024b) as the base model. The base model remains frozen throughout all experiments, and we adopt its default hyperparameters (e.g., noise sampling schedule, conditional guidance scale). Each generated video is 5 seconds long, consisting of 49 frames at a resolution of $720 \times 480$. For *text-based control*, the learning rate for LoRA optimization is set to 1e-4 for appearance learning and 3e-4 for camera motion learning, with the LoRA rank fixed at 128. We use one camera LoRA for different motion types, each using 500 synthetic training videos. For *trajectory-based control*, we fine-tune from the pre-trained `AC3D` model using a learning rate of 1e-4. We train one encoder for different motion types, using the same set of synthetic training videos as in text-based control.

To help reproduce our results, we report the detailed hyperparameter settings in Table 6.

## D  QUALITATIVE COMPARISON AND ANALYSIS

Section 5.2 presents the qualitative results of the text-based methods (VIVIDCAM-COG). In this section, we first present the qualitative results of the trajectory-based methods (VIVIDCAM-AC3D), followed by a comparison with the baseline methods and corresponding analyses.

**Qualitative results of trajectory-based methods.** We present the qualitative results of VIVIDCAM-AC3D in Figure 6. As shown, similar to the text-based method, our trajectory-based method can generate videos with precise camera control and high visual quality across a range of camera motions, from simple and composed to complex ones.

**Qualitative comparison with baselines.** We present qualitative comparisons in Figure 7 and Figure 8. Our observations indicate that state-of-the-art methods struggle to accurately synthesize

Figure 6: Qualitative results of diverse camera motions using VIVIDCAM-AC3D. Note that some complex camera motions are difficult to demonstrate through images; please refer to the videos on our anonymous webpage `https://anonymoususers196.github.io/VividCamDemo/` for better visual results.

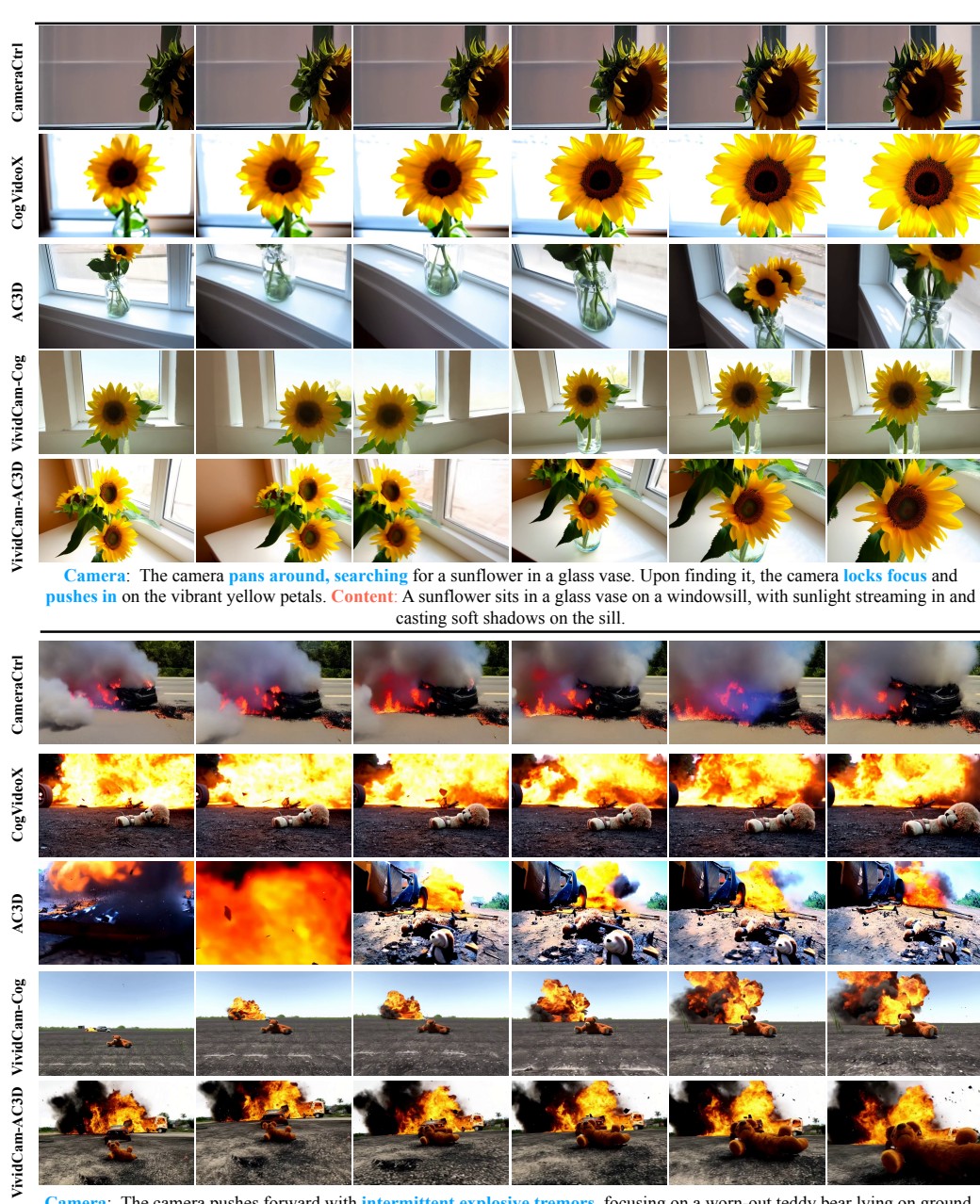

**Camera**: The camera **pans around, searching** for a sunflower in a glass vase. Upon finding it, the camera **locks focus** and **pushes in** on the vibrant yellow petals. **Content**: A sunflower sits in a glass vase on a windowsill, with sunlight streaming in and casting soft shadows on the sill.

**Camera**: The camera pushes forward with **intermittent explosive tremors**, focusing on a worn-out teddy bear lying on ground. **Content**: A car explodes in a fiery blast, sending debris and dust into the air.

Figure 7: Qualitative results comparison. We observe that camera motions such as "panning around to search for an object, then pushing in to focus on the object" are particularly challenging for state-of-the-art models. Even when provided with exact trajectories, these methods often degrade into simpler camera motions—such as a rightward truck in CAMERACTRL or a turbulent push-in in AC3D. In contrast, our method faithfully produces the intended camera motions. Additionally, we note that certain effects, such as explosive camera motions, are difficult to convey through static images. Please refer to the videos on our anonymous webpage `https://anonymoususers196.github.io/VividCamDemo/` for better visual results.

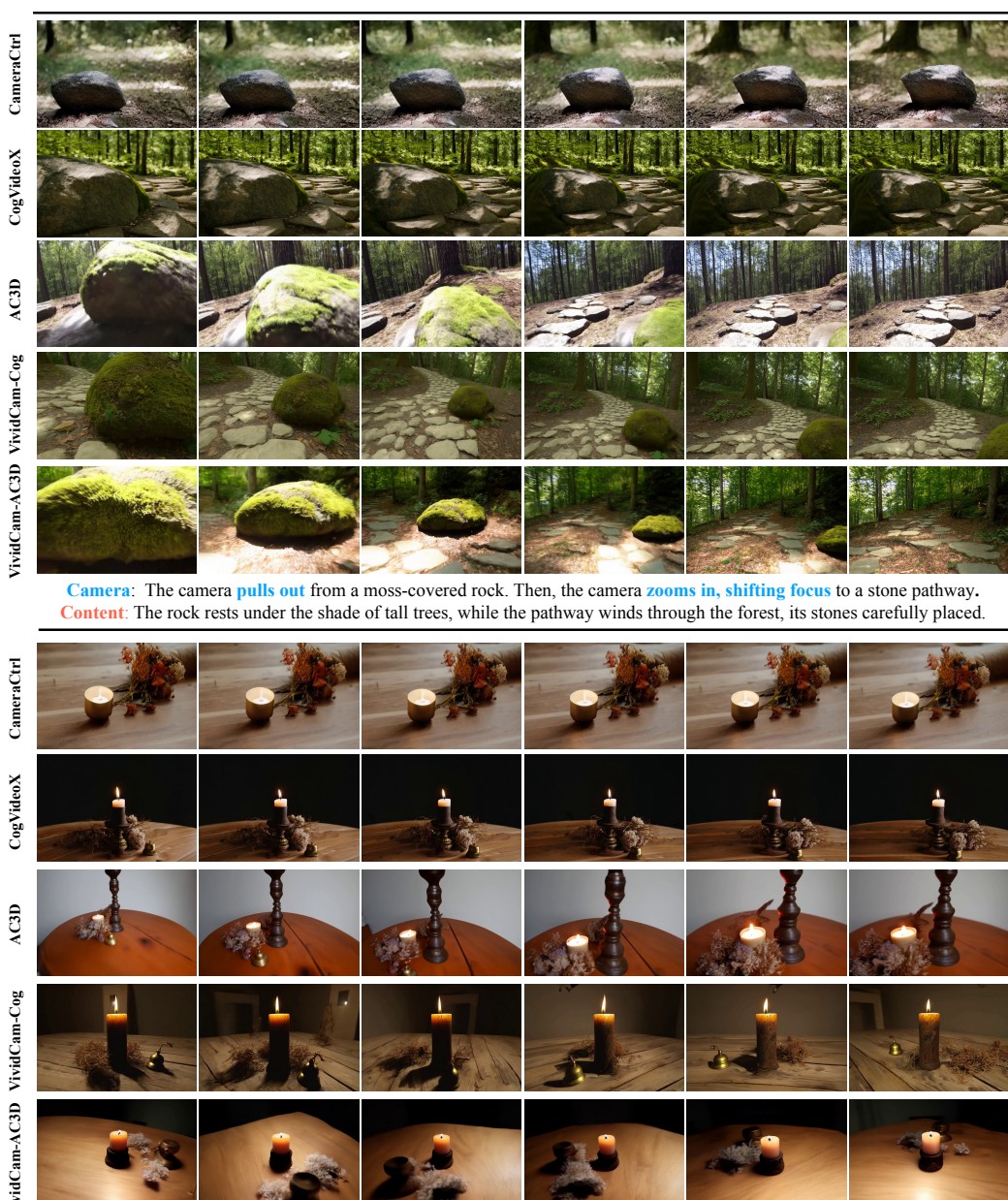

**Camera**: The camera **pulls out** from a moss-covered rock. Then, the camera **zooms in, shifting focus** to a stone pathway.
**Content**: The rock rests under the shade of tall trees, while the pathway winds through the forest, its stones carefully placed.

**Camera**: The camera **orbits** the static candle holder.
**Content**: A rustic candle holder with a flickering candle rests on a wooden table, accompanied by dried flowers and a brass bell.

Figure 8: Qualitative results comparison. We observe that camera motions such as "pulling out from an object, then zooming in to shift focus to another object" are particularly challenging for state-of-the-art models. Even when provided with exact trajectories, these methods often fail to accurately reproduce the desired camera motions. For example, while AC3D attempts to depict a focus shift from a rock to a stone, it does not successfully demonstrate the pull-out from the rock followed by the push-in toward the road. In contrast, our method faithfully captures and reproduces the intended camera motions. Additionally, we note that such unconventional camera movements are difficult to fully appreciate through static images alone. Please refer to the videos on our anonymous webpage https://anonymoususers196.github.io/VividCamDemo/ for better visual results.

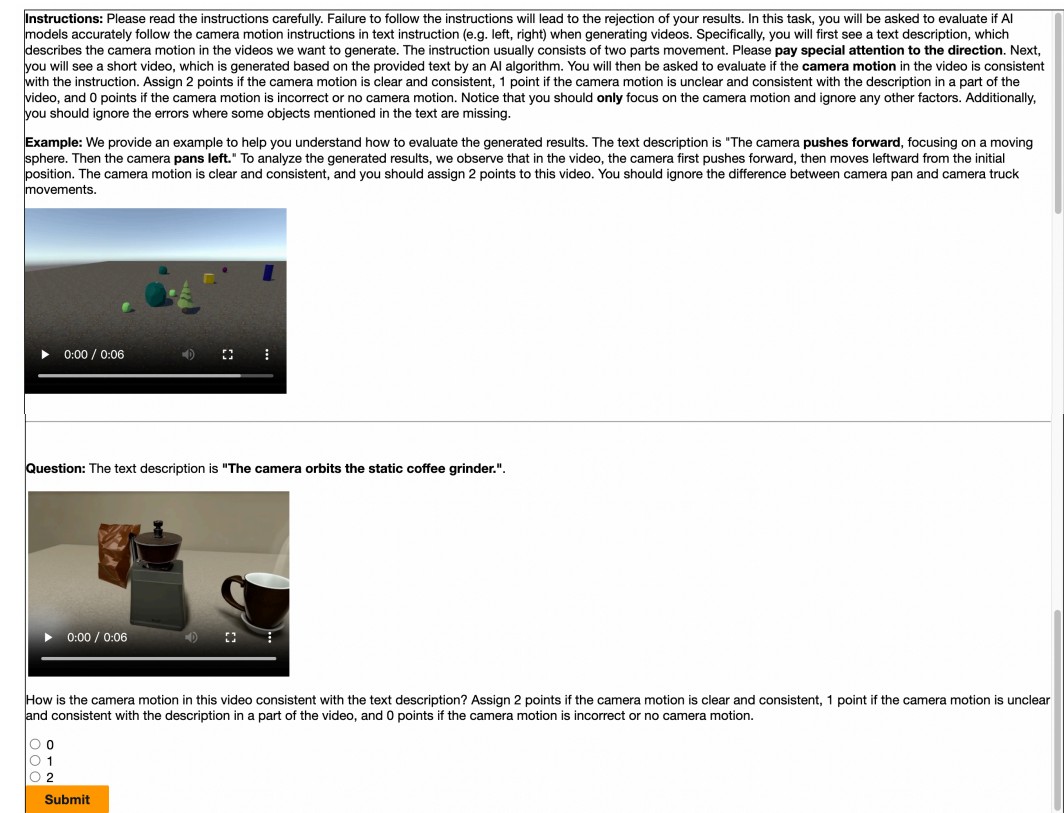

Figure 9: The example interface of Amazon Mechinical Turk in our human study.

unconventional camera motions. For instance, in the upper panel of Figure 7, even when provided with exact trajectories, these methods often simplify the intended motion—resulting in a pan to the right in CAMERACTRL or a turbulent push-in in AC3D. Similarly, in the upper panel of Figure 8, while AC3D attempts to depict a focus shift from a rock to a stone, it fails to effectively illustrate the pull-out from the rock followed by a push-in toward the road. In contrast, our method faithfully captures and reproduces the intended camera motions. Additionally, we note that such unconventional motions are difficult to fully appreciate through static images alone. We encourage readers to refer to the videos on our anonymous webpage `https://anonymoususers196.github.io/VividCamDemo/` for better visual results.

## E    DETAILS OF HUMAN STUDY

Our human study is conducted on Amazon Mechanical Turk. We consider three levels of camera motion: *simple*, *composed*, and *complex*. Please refer to Table 1 for the specific camera motions covered in each category. For each category, we sample 25 prompts and input them into our model and baseline models for evaluation. Each of the 25 prompts is tested twice, resulting in 50 videos per camera motion category and a total of 150 videos. Each question is awarded $0.03. In total, 88 unique workers participate in the study. For each question, we present the tested videos along with the input text prompt and ask participants to answer two types of questions: ❶ (**Action Correctness**) How consistent is the camera motion in the video with the text description? ❷ (**Realism**) How is the visual quality of the video? Participants rate each question on a scale from 0 to 2. To ensure precise evaluation, we provide detailed explanations, a scoring rubric, and examples.

Figure 9 shows an example of the interface that participants will see during the human study.

## F    CLIP SIMILARITY

We present the CLIP similarity results in Table 7. Overall, all methods achieve comparable CLIP similarity scores. Specifically, both VIVIDCAM-COG and VIVIDCAM-AC3D exhibit less than a 0.01 difference in CLIP score compared to their vanilla counterparts, COGVIDEOX and AC3D, respectively. This indicates that our methods

|  | Simple | Composed | Complex |
|---|---|---|---|
| CAMERACTRL | 0.3254 | 0.3306 | 0.3006 |
| COGVIDEOX | 0.3272 | 0.3215 | 0.3070 |
| AC3D | 0.3397 | 0.3437 | 0.3153 |
| VIVIDCAM-COG | 0.3327 | 0.3362 | 0.3230 |
| VIVIDCAM-AC3D | 0.3352 | 0.3341 | 0.3134 |

Table 7: Results on CLIP similarity.

maintain the same level of alignment with the desired content described in the text prompts. We also observe that COGVIDEOX performs worse in scenarios involving complex camera motions, possibly due to quality degradation when generating motion patterns that are likely underrepresented in the training dataset, as shown in Sec. D.

