# OpenReview forum: "VividCam: Learning Unconventional Camera Motions from Virtual Synthetic Videos"
_ICLR.cc/2026/Conference — Submitted to ICLR 2026_

### Official Review · Reviewer_4EJL · 2025-10-27

**Soundness:** 2
**Presentation:** 3
**Contribution:** 2
**Rating:** 2
**Confidence:** 5

**Summary:**

This paper proposes VividCam, a framework for training video diffusion models to generate unconventional camera motions using only synthetic training data rendered from simple low-poly 3D scenes in Unity. The method employs a dual-adaptation training scheme with appearance and camera LoRAs, optical flow loss, and style-aligned prompts to disentangle synthetic appearance from camera motion learning.

**Strengths:**

1. Good qualitative results showing diverse motions
2. Both text-based and trajectory-based control variants
3. Ablation studies on key components

**Weaknesses:**

1. Only tested on one base model (CogVideoX-5B).
2. No failure case analysis
3. Baselines (CameraCtrl, AC3D) weren't trained on the same synthetic data, making comparison unfair
4. Besides the camera control metrics (TransErr, RotErr), there only one appearance-related metric FVD, which is not enough, more metrics from different aspects are needed, for example the metric used to measure the dynamic degree.
5. The main technique innovations, LoRA tuning to avoid appearance leaky is widely used in many different image / video generation methods.
6. The dataset only focus on single domain, leading to low generalization capacities when used to generate other domain's videos.

**Questions:**

1. Since the CogVidX and VIVIDCAM-COG are text based camera control models, how to measure the transerr and roterr in Table 1?
2. Why only use 500 real and 500 synthetic videos for ablating the effectiveness of joint training?
3. What does the validation set consist of?
4. Why does Table 5 show mixed data performs worse than virtual-only data? This contradicts intuition and needs investigation.

---

> ### Author Response · Authors · 2025-11-27
>
> We thank reviewer 4EJL for the thoughtful review. Some of the questions you raised require training on additional foundation model, which takes extra time to complete. We provide detailed responses to each of your questions below.
>
> **Q1:** Only tested on one base model (CogVideoX-5B).
>
> **A1:** Thank you for the suggestion. Due to limited computational resources, we were not able to experiment with various base models at the time of submission. Based on your feedback, we conduct two additional experiments:
> * Training on Wan-AI/Wan2.2-TI2V-5B, a base model that is architecturally different from CogVideoX.
> * Training on Cseti/CogVideoX-LoRA-Wallace_and_Gromit, a LoRA-fine-tuned variant of CogVideoX-5B that focuses on a cartoon domain.
>
> In both settings, VividCam is still able to generate unconventional camera motions. We have added the resulting videos to the anonymous web link (at the end of the page). These results suggest that VividCam can generalize across different base models and across different domains.
>
> **Q2:** No failure case analysis.
>
> **A2:** Thank you for your suggestions. We conducted a careful analysis of the negative examples. As summarized in the table below, we categorize them into two types. The first category includes long semantic motions, which involve camera movements with specific meanings that require more time to be fully conveyed. This type is challenging because the base model, CogVideoX, is limited to 49 frames, making it difficult to express these longer motions within the available duration. The second category consists of drastic movements, where the camera shifts or rotates very rapidly between frames. These sudden changes make it harder to keep the scene consistent and produce smooth outputs. We will update the paper with a section that discusses these two categories.
>
> | Challenging Categories | Long Semantic Motions                                                                                                                                   | Motions with Drastic Movements                                                                          |
> |------------------------|---------------------------------------------------------------------------------------------------------------------------------------------------------|---------------------------------------------------------------------------------------------------------|
> | Examples               | Sweeping across three objects, pausing briefly at each to mimic human examination  | Orbiting the subject with 360 degrees or more  |
> | | Street-wide panning shot, gradually revealing shops and surroundings | Rapid switches between objects with abrupt camera shifts |
>
> **Q3:** Baselines (CameraCtrl, AC3D) weren't trained on the same synthetic data, making comparison unfair.
>
> **A3:** Thank you for the comment. We did not train the baselines on the synthetic data because the synthetic videos are visually and perceptually very different from real videos, as shown in Figure 3. Training the baselines on such data leads to a noticeable drop in visual quality. To illustrate this, we added an experiment where each baseline is trained on exactly the same synthetic videos used for our method. The results, shown in the table below, confirm that training directly on our synthetic data causes a significant reduction in visual quality.
>
> | Setting                                 | TransErr | RotErr | FVD     |
> |-----------------------------------------|----------|--------|---------|
> | VividCAM-Cog                            | 0.4011   | 0.5013 | 1866.40 |
> | VividCam-AC3D                           | 0.3376   | 0.3619 | 1721.45 |
> | CogVideoX                               | 0.4327   | 0.5067 | 1488.36 |
> | CogVideoX trained with synthetic videos | 0.3937   | 0.5224 | 2805.18 |
> | AC3D                                    | 0.4271   | 0.5864 | 1719.82 |
> | AC3D trained with synthetic videos      | 0.4016   | 0.5358 | 2971.21 |

---

> ### Author Response · Authors · 2025-11-27
>
> **Q4:** Besides the camera control metrics (TransErr, RotErr), there is only one appearance-related metric FVD, which is not enough, more metrics from different aspects are needed, for example the metric used to measure the dynamic degree.
>
> **A4:** Thank you for the suggestion. We agree that additional appearance-related metrics can help assess video quality from different perspectives. Following your recommendation, we report the dynamic-degree metric from [1], which is evaluated using a model specifically trained for video assessment. The results are shown below. We find that VividCam achieves higher dynamic-degree scores than its corresponding baselines, indicating its ability to generate more complex and expressive camera motions. We will also add qualitative case studies in the revised paper to illustrate how different types of camera motions influence the dynamic-degree metric.
>
> [1] VideoScore: Building Automatic Metrics to Simulate Fine-grained Human Feedback for Video Generation
>
> | Setting                              | Dynamic Degree [1] |
> |--------------------------------------|----------|
> | VividCAM-Cog                         | 2.781    |
> | - CogVideoX (Corresponding base model) | 2.422    |
> | VividCam-AC3D                        | 3.297    |
> | - AC3D (Corresponding base model)      | 3.187    |
> | CameraCtrl                           | 2.924    |
>
> **Q5:** The main technique innovations, LoRA tuning to avoid appearance leaky, is widely used in many different image / video generation methods.
>
> **A5:** Thank you for the comment. We would like to clarify that the central contribution of our work is not the dual LoRA itself. Our goal is to build an effective pipeline that can generate unconventional camera motions from low-quality synthetic videos by addressing the significant domain gap between these synthetic training videos and the realistic videos produced by modern text-to-video models.
>
> To understand what actually enables this synthetic-to-real transfer, we systematically study several factors that influence the outcome. These include the choice of training videos, the dual-adaptation scheme, the style-aligned prompt, and the motion-consistency loss. We do not claim novelty for these components individually. Instead, the contribution lies in identifying which elements are essential, and how they work together to resolve the synthetic-to-real appearance gap.
>
> Our initial ablations in Table 4 and Figure 5 evaluate the roles of dual-adaptation and the style-aligned prompt. To provide a more complete picture, we additionally examine the effects of the training-data design and the motion-consistency loss (below). These experiments confirm that removing any of the key factors leads to noticeable degradation in realism or motion accuracy.
>
> To the best of our knowledge, prior work has not analyzed how these components behave under such a large synthetic-to-real domain mismatch, nor provided guidance on how to combine them into a practical pipeline. We will revise the manuscript to present this contribution more explicitly.
>
>
> | Setting                             | TransErr | RotErr | FVD     |
> |-------------------------------------|----------|--------|---------|
> | VividCAM-Cog                        | 0.4011   | 0.5013 | 1866.40 |
> | Removing data without camera motion | 0.4305   | 0.4816 | 2921.83 |
> | Removing motion consistency loss    | 0.5291   | 0.5742 | 1911.41 |
>
> **Q6:** The dataset only focuses on a single domain, leading to low generalization capacities when used to generate other domain's videos.
>
> **A6:** Thank you for the suggestion. Based on your feedback, we conducted additional experiments on a cartoon-style domain. For this, we used the base model checkpoint Cseti/CogVideoX-LoRA-Wallace_and_Gromit and trained VividCam on it. We have added the corresponding videos to the anonymous web link. From these results, together with the earlier experiments on realistic videos, we observe that VividCam can generalize to both realistic and cartoon domains. We plan to extend our study to more diverse domains to further demonstrate the generalization capability.

---

> ### Author Response · Authors · 2025-11-27
>
> **Q7:** Since the CogVidX and VIVIDCAM-COG are text based camera control models, how to measure the transerr and roterr in Table 1?
>
> **A7:** Thanks for the great question. When evaluating text-based camera control models such as CogVideoX and VividCam-Cog, we compute TransErr and RotErr by using the corresponding camera motion trajectory of each category as the reference. As noted in the paper (L421–L424), this inherently gives an advantage to trajectory-based methods (both baseline AC3D and ours VividCam-AC3D), since text-based models do not have access to the ground-truth trajectories during inference. To address this limitation, we also conduct human studies to assess whether the generated camera motions are perceived as correct. These human studies avoid unnecessary constraints that require the model to match the reference trajectory exactly, and instead evaluate whether the intended camera motion is conveyed. We will clarify this writing more explicitly in the paper.
>
> **Q8:** Why only use 500 real and 500 synthetic videos for ablating the effectiveness of joint training?
>
> **A8:** Thank you for the comment. In our original ablation study, we used 500 real and 500 synthetic videos to focus on a balanced mixed dataset. We agree that examining larger datasets can provide additional insights. Therefore, we conducted a larger-scale ablation: the virtual-only dataset contains 2,000 synthetic videos, and the mixed dataset contains 2,000 real and synthetic videos. Due to time constraints, we report automated metrics instead of human studies, and the results are shown below. Consistent with Table 5, we find that adding more realistic videos provides minor improvements in realism (FVD), but reduces action correctness. We believe this is because real-world videos rarely contain complex or diverse camera motions, making them less effective for training models that must handle such motions. We further discuss this in [A10].
>
> | Setting      | TransErr | RotErr | FVD     |
> |--------------|----------|--------|---------|
> | Virtual Data | 0.4113   | 0.4927 | 1950.12 |
> | Mixed Data   | 0.5166	   | 0.5048 | 1906.30 |
>
> **Q9:** What does the validation set consist of?
>
> **A9:** Thank you for the question. As discussed in L417–L418, the validation set covers all simple, composed, and complex camera motions. For each category, we include 100 prompts that do not appear in the training data. We will revise the paper to clarify the validation setup and add examples to make it clearer.
>
> **Q10:** Why does Table 5 show mixed data performs worse than virtual-only data? This contradicts intuition and needs investigation.
>
> **A10:** Thank you for the great question. As discussed in [A8], both Table 5 and our additional experiments show that mixed data provides modest improvements in realism. This is likely because the real videos help the model retain the appearance patterns of realistic content during training. However, mixed data performs worse in generating accurate camera motions. After inspecting the data, we find that the RealEstate10K videos have highly similar camera trajectories: they are mostly slow, steady indoor movements such as forward pushes or gentle trucking motions. These patterns differ substantially from the unconventional camera motions we evaluate, such as 180 degree rotations, dolly zooms, and rapid or complex transitions. When mixed data is used, the model tends to overfit to the motion patterns in the real videos and often fails to reproduce these unconventional motions at test time. We will add a case study in the paper to illustrate this behavior more clearly.

---

### Official Review · Reviewer_EfEB · 2025-10-27

**Soundness:** 3
**Presentation:** 4
**Contribution:** 3
**Rating:** 8
**Confidence:** 3

**Summary:**

The paper proposes a method to enhance existing video generation models to support unconventional camera controls. The proposed method first renders synthetic 3D scenes from unity with unconventional camera movements. The rendered scenes are low-poly for easy construction. Then, the paper discussed ways to disentangle the learning of camera movement from the learning of appearance. Specifically, the method first trains a LoRA to only learn the appearance. Then the method learns the camera movement. At inference, the appearance LoRA is dropped to enable unconventional camera control while generating realistic videos. The qualitatively results shows that the method can perform more arbitrary camera movements compared to prior methods such as CameraCtrl. The quantitative metrics also shows that the model has greater alignment with the camera condition.

**Strengths:**

1. The paper is clearly written, with good inspiration for the problem that it tries to tackle. The method description and the evaluation results are presented clearly.

2. The proposed finetuning method is sound and straightforward. The authors also considered the practical difficulty of data curation and decided to use low-poly and simple objects in the rendering pipeline.

3. Evaluation results clearly show the effectiveness of the method. The model can perform more unconventional camera movements compared to prior art.

4. The ablation studies show the effectiveness of the appearance LoRA, as removing the appearance LoRA causes degradation in appearance, justifying the method design.

**Weaknesses:**

1. The method requires the curation of a synthetic dataset. Alternatively, I do believe unconventional/artistic camera movements can be found in massively available gaming videos and movies, which can be extracted and used as conditions following general methods such as CameraCtrl2. It is a trade-off between scaling data and scaling manual work. This is a general weakness of methods requiring synthetic data, not a reflection on the novelty of the method proposed.

2. The authors should consider including related work [1], which also explores fine-tuning video generation models with a synthetic video dataset. Although the work doesn't directly focus on camera control, it has also demonstrated that synthetic videos can be used to support new camera motions.

3. The paper proposed "optical-flow-based" loss in equation 5. First, I think calling it optical-flow-based may be misleading. The loss is more like a temporal derivative consistency loss. But regardless of the naming, which is not a main concern, the paper does not seem to present an ablation study on this proposed loss term.

[1] Synthetic Video Enhances Physical Fidelity in Video Synthesis (ICCV2025)

**Questions:**

1. Regarding weakness 1, although the author has considered using low-poly to simplify the data curation process, the scene still requires textures (sky, ground), and objects, etc. What is the minimum extent that the scene can be? Can it be plain colors? Can it be wireframes?

2. Regarding weakness 3, it would be nice if the author could provide supporting justifications for the use of the loss term.

3. Are you planning on open-sourcing the curated data or synthetic scene generation code?

---

> ### Author Response · Authors · 2025-11-25
>
> We thank reviewer EfEB for the thoughtful review. We appreciate your positive feedback on the clarity of our writing, the straightforward method design, and the strong experimental results.
>
> **Q1:** The method requires the curation of a synthetic dataset. Alternatively, I do believe unconventional/artistic camera movements can be found in massively available gaming videos and movies, which can be extracted and used as conditions following general methods such as CameraCtrl2. It is a trade-off between scaling data and scaling manual work. This is a general weakness of methods requiring synthetic data, not a reflection on the novelty of the method proposed.
>
> **A1:** Thank you for the comment. We agree that unconventional or artistic camera motions can also be manually sourced from gaming videos or movies, and that both real-world and synthetic data have their own advantages. Our choice of synthetic data is motivated by its ease of creation and scalability. In our pipeline, each training video takes about 5 seconds to synthesize in Unity, and generating the full dataset requires less than 2 hours without any manual annotation. This allows us to define precise 3D trajectories and produce consistent supervision at scale. We see synthetic and real data as complementary research directions, and our focus in this work is to demonstrate that a fully synthetic dataset can effectively support realistic camera-motion generation.
>
> **Q2:** The authors should consider including related work [1], which also explores fine-tuning video generation models with a synthetic video dataset. Although the work doesn't directly focus on camera control, it has also demonstrated that synthetic videos can be used to support new camera motions.
>
> [1] Synthetic Video Enhances Physical Fidelity in Video Synthesis (ICCV2025)
>
> **A2:** Thank you for the suggestion. We have carefully read this work and find that it provides a detailed analysis of how synthetic videos can improve generated video quality, which aligns with the general direction of our research. We will include a detailed discussion of this work in the related work section.
>
> **Q3:** The paper proposed "optical-flow-based" loss in equation 5. First, I think calling it optical-flow-based may be misleading. The loss is more like a temporal derivative consistency loss. But regardless of the naming, which is not a main concern, the paper does not seem to present an ablation study on this proposed loss term.
>
> **A3:** Thank you for the suggestion. We agree that “temporal derivative consistency loss” is a more accurate name for this term, since it reflects how the loss is computed, and we will revise the wording in the paper.
>
> We also appreciate the suggestion to provide an ablation study. We have added this experiment, and the results are shown below. The ablation indicates that this temporal loss plays an important role in producing precise camera motions.
>
> | Setting                          | TransErr | RotErr | FVD     |
> |----------------------------------|----------|--------|---------|
> | VividCAM-Cog                     | 0.4011   | 0.5013 | 1866.40 |
> | Removing optical-flow based loss | 0.5291   | 0.5742 | 1911.41 |
>
> **Q4:** Regarding weakness 1, although the author has considered using low-poly to simplify the data curation process, the scene still requires textures (sky, ground), and objects, etc. What is the minimum extent that the scene can be? Can it be plain colors? Can it be wireframes?
>
> **A4:** Thank you for raising this question. We find that textures and a variety of objects both influence the quality of the generated videos. Based on the results below, diverse object categories help improve realism in the final outputs, and textures support both correct motion and realistic appearance. We find this is because having different objects prevents the model from overfitting to specific shapes or wireframes, and textures provide visual cues for relative motion. Without textures on the objects and the environment, it becomes difficult to tell the camera movement.
>
> However, we also note that adding various object templates and specifying textures is a one-time setup in Unity that requires only a few lines of code, and it does not introduce any additional manual effort for generating each synthetic video.
>
> | Setting                                                      | TransErr | RotErr  | FVD      |
> |--------------------------------------------------------------|----------|---------|----------|
> | VividCAM-Cog                                                 | 0.4011   | 0.5013  | 1866.40  |
> | Limiting landscape and object variety to (2 categories each) | 0.4014   | 0.5247  | 2261.91  |
> | Plain color objects/landscape without any texture            | 0.4929   | 0.5603  | 2184.31  |
> | Wireframe objects                                            | 0.4263   | 0.4955  |  2719.06 |

---

> > ### Comment · Reviewer_EfEB · 2025-11-25
> >
> > Thank you for addressing my questions and I have no further questions. I am still convinced that the work is above the acceptance standard after considering other reviewer's comments and the authors' rebuttal. I will keep my original score unchanged.

---

> > > ### Author Response · Authors · 2025-11-27
> > >
> > > Thank you for reviewing our rebuttal. We are encouraged by your positive feedback, and we sincerely appreciate your time and effort.

---

> ### Author Response · Authors · 2025-11-25
>
> **Q5:** Regarding weakness 3, it would be nice if the author could provide supporting justifications for the use of the loss term.
>
> **A5:** Thank you for the comment. Please refer to [A3] for the ablation study of the loss term.
>
> **Q6:** Are you planning on open-sourcing the curated data or synthetic scene generation code?
>
> **A6:** Thank you for the comment. The code for data creation, synthetic scene generation, and model training is currently under internal review. However, the code for creating the synthetic scenes is straightforward, and we provide a detailed description of the setup and the required Unity scripts in Appendix A. This allows others to reproduce the dataset without requiring extensive Unity expertise.

---

### Official Review · Reviewer_6Ycn · 2025-10-28

**Soundness:** 3
**Presentation:** 3
**Contribution:** 2
**Rating:** 4
**Confidence:** 4

**Summary:**

The paper proposes VIVIDCAM, which uses synthetic video with diverse camera motion to fine-tune video generation models to generate camera motion. With several disentanglement strategies, VIVIDCAM can learn robust motion representation from synthetic videos. Experiments show that the proposed method could generate real videos with unconventional motions.

**Strengths:**

1. The proposed method provides an efficient way to enable video generation models to generate diverse camera motions.
2. Extensive experiments are conducted to show the effectiveness of the proposed method.

**Weaknesses:**

1. There are four techniques for disentanglement, ie, dual-adaptation training, data with and without camera motion, optical-flow based loss, and special text prompt. However, the contribution of each one is not well present in the paper, although the ablation has explored two of them.
2. The training setting of the comparing methods and the proposed method seems to be different. Are the comparing method trained on the same data?
3. The training and testing combination of motion is unclear. What are the motion combinations from Table 1 in testing? Are these combinations seen in the training? Can the methods generalize to unseen combinations or even unseen motion (not defined in Table 1)?
4. The novelty of the proposed method is limited. It combines four techniques for disentanglement but doesn't provide new insights or methods for this task.

**Questions:**

1. What is the individual contribution of the techniques for disentanglement?
2. What is the training setting of the proposed method?
3. Waht are the motion combinations in the training and testing?

---

> ### Author Response · Authors · 2025-11-25
>
> We thank reviewer 6Ycn for the thoughtful review.
>
> **[Weakness 1, Q1]** There are four techniques for disentanglement, ie, dual-adaptation training, data with and without camera motion, optical-flow based loss, and special text prompt. However, the contribution of each one is not well present in the paper, although the ablation has explored two of them.
>
> **A1:** Thank you for the suggestions. In addition to the ablation studies on dual-adaptation and the text prompt, we have added ablations to examine the effects of the training data and the optical-flow based loss. The results are summarized below.
>
> * The role of training data: We did not include this ablation initially because the dual-adaptation scheme requires two categories of data. For this study, we modified the training data so that all videos contain camera motion. This leads to a significant degradation in visual quality, as reflected by the increased FVD scores. To further analyze the contribution of the training video design, we also ablated the diversity of landscapes and objects. These experiments show that these design choices are important; without them, the generated videos appear noticeably less realistic.
>
> | Setting                                                      | TransErr | RotErr | FVD     |
> |--------------------------------------------------------------|----------|--------|---------|
> | VividCAM-Cog                                                 | 0.4011   | 0.5013 | 1866.40 |
> | Removing data without camera motion                          | 0.4305   | 0.4816 | 2921.83 |
> | Limiting landscape and object variety to (2 categories each) | 0.4014   | 0.5247 | 2261.91 |
>
> * The role of optical-flow based loss: Following your suggestion, we added an ablation for the optical-flow loss. The results indicate that this loss plays an important role in producing precise camera motions.
>
> | Setting                          | TransErr | RotErr | FVD     |
> |----------------------------------|----------|--------|---------|
> | VividCAM-Cog                     | 0.4011   | 0.5013 | 1866.40 |
> | Removing optical-flow based loss | 0.5291   | 0.5742 | 1911.41 |
>
> **[Weakness 2, Q2]:** The training setting of the comparing methods and the proposed method seems to be different. Are the comparing method trained on the same data?
>
> **A2:** Thank you for the comment. In our experiments, the baselines are not trained on the synthetic data. The synthetic videos are visually and perceptually very different from real videos, as shown in Figure 3, and training the baselines on them leads to a noticeable drop in visual quality. To illustrate this, we added an experiment where each baseline is trained on exactly the same synthetic videos used for our method. The results are shown in the table below and confirm that training directly on our synthetic data causes a significant reduction in visual quality.
>
> | Setting                                 | TransErr | RotErr | FVD     |
> |-----------------------------------------|----------|--------|---------|
> | VividCAM-Cog                            | 0.4011   | 0.5013 | 1866.40 |
> | VividCam-AC3D                           | 0.3376   | 0.3619 | 1721.45 |
> | CogVideoX                               | 0.4327   | 0.5067 | 1488.36 |
> | - CogVideoX trained with synthetic videos | 0.3937   | 0.5224 | 2805.18 |
> | AC3D                                    | 0.4271   | 0.5864 | 1719.82 |
> | - AC3D trained with synthetic videos      | 0.4016   | 0.5358 | 2971.21 |
>
> **[Weakness 3, Q3]:** The training and testing combination of motion is unclear. What are the motion combinations from Table 1 in testing? Are these combinations seen in the training? Can the methods generalize to unseen combinations or even unseen motion (not defined in Table 1)?
>
> **A3:** Thank you for the comment. The motions in Table 1 are included in both training and testing. The model does not generalize reliably to unseen motions because different camera motions correspond to distinct transformations in 3D space. For example, rotating the camera upside-down involves a large 3D rotation, while seeking an object is dominated by translational movement. These motion types differ structurally, so learning one does not provide supervision for another.
>
> Although the model cannot infer entirely new motion categories on its own, our pipeline supports adding new camera motions efficiently. Each motion can be defined by a new 3D trajectory in Unity, and synthesizing the corresponding training videos takes only a short time (about 5 seconds per video) in our setup. This makes it straightforward to extend the motion set if new motions are desired.

---

> ### Author Response · Authors · 2025-11-25
>
> **[Weakness 4]:** The novelty of the proposed method is limited. It combines four techniques for disentanglement but doesn't provide new insights or methods for this task.
>
> **A4:** Thank you for the comment. We would like to clarify that the contribution of our work is not the introduction of new architectural components. Our goal is to build an effective pipeline to generate unconventional camera motions from low-quality synthetic videos by addressing the significant domain gap between these synthetic training videos and the realistic videos produced by modern text-to-video models.
>
> To understand what actually enables this synthetic-to-real transfer, we systematically study several factors that influence the outcome: the choice of training videos, the dual-adaptation scheme, the style-aligned prompt, and the motion-consistency loss. We do not claim novelty for any of these components individually. Instead, the contribution lies in identifying which of these elements are essential, and how they collectively resolve the synthetic-to-real appearance gap. Our ablations (Table 4, Figure 5, and the supplemental experiments added in response to your suggestions in [A1]) show that omitting any of the key components leads to noticeable degradation in realism or motion accuracy.
>
> To the best of our knowledge, prior work has not analyzed how these factors behave under such an extreme synthetic-to-real domain mismatch, nor provided guidance on how to combine them to make synthetic videos usable for realistic camera-motion generation. We will revise the manuscript to present this contribution more explicitly.

---

> > ### Comment · Reviewer_6Ycn · 2025-11-26
> >
> > Thank you for providing new results. Most of my concerns are addressed in the rebuttal. I have updated my rating.

---

> > > ### Author Response · Authors · 2025-11-27
> > >
> > > Thank you for reviewing our rebuttal and updating your rating. We are happy to discuss any further concerns you may have. We sincerely appreciate your time and effort.

---

### Official Review · Reviewer_5sQc · 2025-10-31

**Soundness:** 2
**Presentation:** 2
**Contribution:** 1
**Rating:** 4
**Confidence:** 4

**Summary:**

This paper proposes VIVIDCAM, a method for training text-to-video diffusion models to follow specific camera motions, particularly "unconventional" ones for which real-world training data is scarce. The core idea is to use low-poly, synthetic videos (rendered in Unity) as the training data.

To prevent the synthetic, "low-poly" appearance from bleeding into the final output, the authors propose a dual-adaptation training strategy. First, an "appearance LoRA" is trained on static synthetic videos to learn and isolate the synthetic visual style. This training is guided by a special "virtual indicator" tag in the prompt. Second, a camera motion module (either a LoRA for text-based control or fine-tuning a trajectory encoder) is trained on synthetic videos that do contain camera motion.

At inference time, the appearance LoRA is discarded, and the model is expected to produce realistic videos that follow the camera motions learned from the synthetic data. The method is evaluated on text-based and trajectory-based camera control, using automated metrics (FVD, TransErr) and human studies.

**Strengths:**

The problem itself is significant. Enabling controllable, complex, and artistic camera motions is a key challenge for creative video generation, and the lack of diverse, well-labeled real-world data is a real bottleneck.

The idea of using synthetic data is a logical approach to solving this data scarcity problem.

**Weaknesses:**

The paper suffers from a significant lack of novelty, a mismatch between its tools and goals, and an unconvincing evaluation.

1. Critical Lack of Novelty: The central contribution, the "dual adaptation" or "dual LoRA" method for disentangling appearance from motion, is not new. This exact training pattern was established by AnimateDiff (Guo et al., 2023), which the authors cite as "inspiration." AnimateDiff's "Stage 1: Domain Adapter" is functionally identical to this paper's "Step 1: Appearance Adaptation." VIVIDCAM is a direct application of this existing 2-stage (adapter-then-motion) training strategy to a synthetic dataset. This is an incremental extension, not a novel framework, and represents a major overstatement of the paper's contribution.

2. Mismatch of Tool and Goal: The paper uses a powerful 3D rendering engine (Unity) capable of generating perfect, precise, 6-DOF camera trajectories (i.e., extrinsics). However, a large part of the paper focuses on using this data to train a model on vague, ambiguous text prompts like "pan left" or "push in." This is a baffling waste of the synthetic data's primary advantage. It's a step backward from existing works (including its own baselines like AC3D and CameraCtrl) that are focused on fine-grained, trajectory-based control.

3. Unconvincing Rationale: The paper motivates the work by the need for "unconventional" motions. However, the motions listed in Table 1 ("Simple" and "Composed") are entirely conventional (push, tilt, truck). These are readily available in existing real-world datasets. While the "Complex" motions (e.g., "seek object") are more interesting, training them from ambiguous text prompts is far less compelling than training a model to follow an explicit 3D "seeking" path, which Unity could have easily generated.

4. Insufficient Evaluation & Unconvincing Ablations: The primary evidence for the core disentanglement claim rests on the ablation in Figure 5. This figure not only fails to show temporal artifacts (by only showing static frames), but the visual difference between the full model and the "w/o Style-aligned Prompts" version is minimal. This does not convincingly demonstrate that the appearance LoRA and [VIRTUAL] tag are doing the heavy lifting the authors claim. Furthermore, the paper lacks direct side-by-side video comparisons to its baselines, making it impossible to truly judge the trade-off between motion accuracy and visual fidelity.

5. Poor Qualitative Fidelity: When compared to the baselines (e.g., vanilla CogVideoX or AC3D), the visual performance of the proposed method appears to be a significant regression. The generated videos, while perhaps following the motion, are less realistic and have lower fidelity than the state-of-the-art baselines they are built upon. The method seems to trade visual quality for motion control, which is a poor trade-off.

**Questions:**

1. The camera motion module (LoRA/encoder) is also trained with the <VIRTUAL> tag in the prompt. At inference, this tag is removed. How do you know the motion module isn't confused or degraded, as it was never trained on prompts without this tag? A key ablation is missing: what happens to motion accuracy if you keep the tag at inference (accepting the synthetic output)?

2. Given that AnimateDiff's "Domain Adapter" + "Motion Module" training pipeline is functionally identical to this paper's, could the authors please state clearly what the technical novelty of this paper is, beyond just applying this pattern to a new synthetic dataset?

3. Why was the decision made to degrade the perfect 3D trajectory data from Unity into coarse text prompts? Why not use the synthetic data for its primary strength and train a model that takes explicit 3D paths, target-of-interest coordinates, or specific orbit/dolly-zoom parameters as input?

4. Table 1 lists "Dolly zoom" as a capability. A true Dolly zoom has a specific mathematical definition (simultaneously changing focal length and camera distance to keep the subject size constant). Can the text-based model actually reproduce this effect from a simple prompt, or is it just generating a standard "zoom in"?

5. Could the authors provide a more in-depth analysis of why the appearance LoRA, trained only with a [VIRTUAL] tag, is sufficient to "absorb" all synthetic artifacts? The mechanism seems very similar to DreamBooth (specializing to a token) but is used for a different purpose (disentanglement), and this is not well-justified.

---

> ### Author Response · Authors · 2025-11-25
>
> We thank reviewer 5sQc for the thoughtful review.
>
> **Q1:** Critical Lack of Novelty: The central contribution, the "dual adaptation" or "dual LoRA" method for disentangling appearance from motion, is not new. This exact training pattern was established by AnimateDiff (Guo et al., 2023).
>
> **A1:** Thank you for the comment. We would like to clarify that the central contribution of our work is **not** the dual LoRA itself. Our main goal is to build an effective pipeline to generate unconventional camera motions from low-quality synthetic videos, addressing the significant domain gap between the synthetic training videos and the generated realistic videos (as described in L116 - 119).
>
> AnimateDiff appearance adaptation uses a removable domain adapter to reduce the quality discrepancy between video frames and high-quality image data. Its purpose is to absorb issues such as motion blur and compression artifacts in videos. It is not intended to address the appearance shift between synthetic training videos and realistic generated videos. In fact, we find simply applying dual LoRA alone does not resolve this gap. Table 4 and Figure 5 show that dual LoRA by itself fails to produce realistic results.
>
> Our method examines several factors that influence whether the synthetic-to-real transfer succeeds. These include the choice of training videos, dual adaptation, style-aligned prompts, and the motion-consistency loss. We do not claim any of these factors as individually novel. The contribution lies in identifying and validating a pipeline that works for this new domain transfer problem. To the best of our knowledge, prior work has not investigated how these factors behave under such an extreme synthetic-to-real appearance gap.
>
> We will revise the wording to make the intended contributions less ambiguous.
>
> **Q2:** Mismatch of Tool and Goal: The paper uses a powerful 3D rendering engine (Unity) capable of generating precise camera trajectories (i.e., extrinsics). However, a large part of the paper focuses on using this data to train a model on vague, ambiguous text prompts like "pan left" or "push in."
>
> **A2:** Thank you for the comment. It seems there may have been some confusion regarding our setting. Our work considers two usage scenarios:
>
> (1) Text-based scenarios, where the user specifies camera motions using plain text, and
>
> (2) **Trajectory-based scenarios**, where the user specifies camera motions using precise camera trajectories (extrinsic camera matrices).
>
> The trajectory-based scenario is designed specifically for precise camera control using the data generated by Unity. As shown in Table 2 and Table 3 (and in the quantitative results in our anonymous link), both scenarios produce high-quality videos. We include both to demonstrate that Unity-generated synthetic videos can support not only accurate trajectory control but also more flexible and ambiguous text instructions, which reflect realistic user needs.
>
> **Q3.1:** Unconvincing Rationale: The paper motivates the work by the need for "unconventional" motions. However, the motions listed in Table 1 ("Simple" and "Composed") are entirely conventional (push, tilt, truck). These are readily available in existing real-world datasets.
>
> **A3.1:** Thank you for the comment. Our work focuses on generating unconventional camera motions, but in the experiments we also include simple and composed motions. This is to demonstrate that our method can handle **both** basic camera operations and more complex or less common ones. Including results on standard motions allows for clearer comparison with baselines and provides a more complete picture of the method's capabilities.
>
> **Q3.2:** While the "Complex" motions (e.g., "seek object") are more interesting, training them from ambiguous text prompts is far less compelling than training a model to follow an explicit 3D "seeking" path, which Unity could have easily generated.
>
> **A3.2:** Thank you for the comment. This scenario is included in our work. As described in **A2**, our **trajectory-based** setting allows the user to provide an explicit 3D path, and the model generates videos that follow this path directly.
>
> **Q4.1:** Insufficient Evaluation & Unconvincing Ablations: The primary evidence for the core disentanglement claim rests on the ablation in Figure 5. This figure not only fails to show temporal artifacts (by only showing static frames), but the visual difference between the full model and the "w/o Style-aligned Prompts" version is minimal.
>
> **A4.1:** Thank you for the comment. It seems there may be some confusions regarding the ablation experiment setting.  We would like to clarify that in addition to Figure 5, we also conduct **quantitative experiments** through human study and report the performance in Table 4. The result demonstrates that without LoRA or style-aligned prompts, the video realisticness is significantly degraded.

---

> ### Author Response · Authors · 2025-11-25
>
> **Q4.2:** The paper lacks direct side-by-side video comparisons to its baselines.
>
> **A4.2:** Thank you for the comment. We would like to clarify that side-by-side video comparisons with all baselines are available in the anonymous link provided in the original submission. These videos show the outputs of both the baselines and our method, and illustrate that the baselines often struggle with complex camera motions.
>
> **Q5:** Poor Qualitative Fidelity: The generated videos, while perhaps following the motion, are less realistic and have lower fidelity than the state-of-the-art baselines they are built upon. The method seems to trade visual quality for motion control, which is a poor trade-off.
>
> **A5:**  Thank you for the comment. The visual quality of generated videos has subjective aspects. To systematically show the visual quality, we report both objective metrics and human studies. Table 2 and Table 3 show that the visual quality of our generated videos is comparable to that of the baselines, based on FVD and user ratings. These results indicate that our method maintains visual quality while enabling camera-motion control, rather than trading one for the other. If there are particular qualitative examples you are concerned about, we are glad to discuss them.
>
> **Q6:** The camera motion module (LoRA/encoder) is also trained with the [object Object] tag in the prompt. At inference, this tag is removed. How do you know the motion module isn't confused or degraded, as it was never trained on prompts without this tag? A key ablation is missing: what happens to motion accuracy if you keep the tag at inference (accepting the synthetic output)?
>
> **A6:** Thank you for the comment. We think you refer to the style-aligned prompt (low-poly Virtual tag) that is dropped at the inference stage. This enables the model to align with virtual style videos at training, and the camera motions are determined by the camera prompts. To validate this, we perform an ablation study on the motion accuracy and the FVD when this tag is also included in the inference. The result shows that including motion accuracy has small effects on the motion accuracy (TransErr and RotErr), but drastically degrades the video quality towards synthetic videos (FVD). This confirms the role of the tag in help aligning the style rather than controlling the camera motions.
>
> | Setting                                             | TransErr | RotErr | FVD     |
> |-----------------------------------------------------|----------|--------|---------|
> | VividCAM-Cog                                        | 0.4011   | 0.5013 | 1866.40 |
> | Including style-aligned prompt tag during inference | 0.4171   | 0.4834 | 2572.08 |
>
> **Q7:** Given that AnimateDiff's "Domain Adapter" + "Motion Module" training pipeline is functionally identical to this paper's, could the authors please state clearly what the technical novelty of this paper is, beyond just applying this pattern to a new synthetic dataset?
>
> **A7:** Thank you. Please refer to [A1] for a complete discussion of the contribution in this work.
>
> **Q8:** Why was the decision made to degrade the perfect 3D trajectory data from Unity into coarse text prompts? Why not use the synthetic data for its primary strength and train a model that takes explicit 3D paths, target-of-interest coordinates, or specific orbit/dolly-zoom parameters as input?
>
> **A8:** Thank you. Our work already supports explicit 3D path inputs. Please refer to [A2] for the full discussion.
>
> **Q9:** Table 1 lists "Dolly zoom" as a capability. A true Dolly zoom has a specific mathematical definition (simultaneously changing focal length and camera distance to keep the subject size constant). Can the text-based model actually reproduce this effect from a simple prompt, or is it just generating a standard "zoom in"?
>
> **A9:** Thank you for the question. We would like to clarify that the dolly-zoom effect can be produced in our text-based setting. As shown in our anonymous link (complex camera motion, row 3, column 1), the generated video changes the focal length, which can be observed from the change in background texture, while keeping the size of the subject (the cup) roughly constant. This behavior matches the definition of a true dolly zoom and is different from a standard zoom-in. We are glad to further discuss if you have concerns regarding the demonstrated dolly zoom effects.

---

> ### Author Response · Authors · 2025-11-25
>
> **Q10:** Could the authors provide a more in-depth analysis of why the appearance LoRA, trained only with a [VIRTUAL] tag, is sufficient to "absorb" all synthetic artifacts? The mechanism seems very similar to DreamBooth (specializing to a token) but is used for a different purpose (disentanglement), and this is not well-justified.
>
> **A10:** Thank you for the great question. The ability to “absorb” synthetic artifacts and still produce realistic videos is not due to the [VIRTUAL] tag alone. In VividCam, several components work together: the two-LoRA adaptation, the design of the synthetic training videos, the dual-adaptation scheme, the style-aligned prompt (tag), and the motion-consistency loss. As shown in Figure 5 and Table 4, removing two-LoRA adaptation or the tag components leads to a clear drop in realism.
>
> In addition, we conducted a more detailed analysis of the remaining components, including the synthetic training data design and the motion-consistency loss. The new experiments (summarized below) show that having diverse synthetic videos also plays an important role in absorbing artifacts and preserving realism.
>
> | Setting                                                            | TransErr | RotErr | FVD     |
> |--------------------------------------------------------------------|----------|--------|---------|
> | VividCAM-Cog                                                       | 0.4011   | 0.5013 | 1866.40 |
> | Data: Limiting landscape and object variety to (2 categories each) | 0.4014   | 0.5247 | 2261.91 |
> | Removing optical-flow based loss                                   | 0.5291   | 0.5742 | 1911.41 |

---

> ### Author Response · Authors · 2025-11-27
> **Follow-up on our rebuttal**
>
> Dear reviewer 5sQc,
>
> Thank you again for your thoughtful and detailed review. As the discussion period progresses, we would greatly appreciate it if you could take a moment to look over our rebuttal. In particular, we hope that the trajectory-based setting described in the paper helps address your main concerns such as “Mismatch of Tool and Goal” and the “Unconvincing Rationale.” We are happy to provide any further clarification or answer additional questions if needed.
>
> Sincerely,
> Authors

---

### Meta-Review · Area_Chair_Q4Mi · 2025-12-27

**Summary:**

The initial reviews raised several critical concerns regarding the technical novelty, the methodological alignment between the synthetic data used and the control goals, and the fairness of the comparative evaluation. Reviewers primarily pointed out that the "dual adaptation" strategy closely mirrors existing works like AnimateDiff, and the focus on ambiguous text-based camera control seems to underutilize the precise 3D data provided by the Unity engine. During the rebuttal, the authors attempted to clarify their contribution as a "pipeline" rather than individual components and provided additional ablations and cross-model tests. However, the fundamental concern remains that the work represents an incremental combination of established techniques (LoRA, style-aligned prompts, and consistency losses) applied to a synthetic-to-real domain transfer problem that was not convincingly shown to be a major technical breakthrough. Furthermore, the qualitative regression in visual fidelity compared to baselines and the lack of generalization to unseen motions persist as significant drawbacks. Therefore, I recommend Reject.

**Reviewer Concerns:**

Addressed during Rebuttal:

- The authors provided a failure case analysis, categorizing issues into long semantic motions and drastic movements, which adds some clarity to the model's limitations.

- Additional quantitative metrics, specifically "dynamic-degree," were included to supplement the initial FVD results.

- Generalization was tested on an additional base model (Wan2.2) and a different domain (cartoon style), suggesting the pipeline's basic functionality is not limited to a single checkpoint.

- The "optical-flow-based" loss was renamed to "temporal derivative consistency loss" to more accurately reflect its function, and a corresponding ablation study was provided.

Outstanding/Major Points:

- Technical Novelty: The core training pattern (dual LoRA for appearance/motion) remains functionally identical to AnimateDiff’s domain adapter strategy. The rebuttal’s defense—characterizing the work as an "effective pipeline"—does not sufficiently elevate the paper above an incremental application of existing ideas.

- Fairness of Baseline Comparison: A major point of contention is that baselines (AC3D, CameraCtrl) were not trained on the same synthetic data. While the authors argue that training baselines on low-quality synthetic data would hurt their performance, this creates an asymmetrical comparison that makes it difficult to isolate whether the gains come from the proposed method or simply the specific data curation.

- Motion Generalization: The model fails to generalize to unseen camera motions. While the authors argue that synthetic data generation is fast, the requirement to retrain or fine-tune for every new motion type limits the practical utility and technical depth of the framework.

- Visual Fidelity vs. Control: Despite the new metrics, some reviewers remain unpersuaded regarding the visual quality trade-off. The qualitative results suggest that the pursuit of camera control leads to a regression in realism compared to the state-of-the-art base models.

**Reviewer Scores:**

Reviewer 5sQc (Rating: 4 $\to$ 4): The reviewer's primary grievance regarding the "critical lack of novelty" and the "mismatch of tool and goal" was not addressed by the rebuttal's re-characterization of the work as a pipeline. The core technical objections remain valid.

Reviewer 6Ycn (Rating: 4 $\to$ 4 or 6): While the added ablations on loss and data partially addressed the reviewer's technical questions, the underlying concern about limited novelty and the lack of insights into the disentanglement task suggests only a marginal, if any, score increase.

Reviewer EfEB (Rating: 8 $\to$ 8): This reviewer was already positive. The authors' agreement to update the loss naming and include related work likely maintains this score, though it stands as an outlier in the overall consensus.

Reviewer 4EJL (Rating: 2 $\to$ 2): The provision of additional model tests and dynamic metrics addresses some procedural weaknesses, but the fundamental critique regarding "technique innovations" and "fairness" in baseline comparisons remains largely unresolved, likely resulting in only a minor upward adjustment.

---

### Decision · Program_Chairs · 2026-01-26

Reject